# RNA fine-tunes estrogen receptor-alpha binding on low-affinity DNA motifs for transcriptional regulation

Deepanshu Soota[1], Bharath Saravanan[1,2,3], Rajat Mann[1,3], Tripti Kharbanda[1] & Dimple Notani [1]✉

## Abstract

**Transcription factors (TFs) regulate gene expression by binding with varying strengths to DNA via their DNA-binding domain. Additionally, some TFs also interact with RNA, which modulates transcription factor binding to chromatin. However, whether RNA-mediated TF binding results in differential transcriptional outcomes remains unknown. In this study, we demonstrate that estrogen receptor α (ERα), a ligand-activated TF, interacts with RNA in a ligand-dependent manner. Defects in RNA binding lead to genome-wide loss of ERα recruitment, particularly at weaker ERα-motifs. Furthermore, ERα mobility in the nucleus increases in the absence of its RNA-binding capacity. Unexpectedly, this increased mobility coincides with robust polymerase loading and transcription of ERα-regulated genes that harbor low-strength motifs. However, highly stable binding of ERα on chromatin negatively impacts ligand-dependent transcription. Collectively, our results suggest that RNA interactions spatially confine ERα on low-affinity sites to fine-tune gene transcription.**

**Keywords** Chromatin; DNA-motifs; Transcription Factors; Non-Coding RNA; Estrogen Receptor-Alpha
**Subject Categories** Cancer; Chromatin, Transcription & Genomics

## Introduction

Transcription factors (TFs) are known for their ability to recognize and bind specific DNA motifs through their DNA Binding Domains (DBD). In addition to DNA binding, a large number of TFs also interact with RNA (Pelham and Brown 1980; Cassiday and Maher 2002; Hudson and Ortlund 2014; Oksuz et al, 2023).

TFs interact with RNA via their DBD, canonical RNA binding motifs (RGG), arginine-rich motifs (ARMs) or low complexity regions (Hendrickson et al, 2016; Mann and Notani 2023). The RNA binding through DBD sequesters TF from binding to its cognate DNA motif (Ahmed et al, 2021; Dickey and Pyle 2017; Hamilton et al, 2022; Holmes et al, 2020; Hudson and Ortlund 2014; Yang et al, 2020). Conversely, RNA interactions via non-DBD regions can modulate the TFs binding dynamics on chromatin that

may lead to transcriptional dysregulation (Clemens et al, 1993; Hou et al, 2020; Oksuz et al, 2023; Saldaña-Meyer et al, 2019; Sigova et al, 2015; Xu and Koenig 2004; Yang et al, 2013; Yoshida et al, 2004). However, the precise function of RNA on chromatin targeting and transcription regulation by TF is poorly understood. Further, the TF-binding motif context and their relative dependence on RNA is poorly understood. Furthermore, does low vs high binding of TF due to RNA provides different transcriptional outcome is unknown.

Estrogen receptor-alpha (ERα) is a ligand-dependent steroid hormone receptor which regulates estrogen induced genes by recruiting polymerase complexes on specific enhancers and promoters (Li et al, 2013). ERα interacts with RNA however, the role of RNA in ERα binding on chromatin and resultant transcription is debated (Oksuz et al, 2023; Steiner et al, 2022; Xu et al, 2021). ERα interacts with RNA either via DBD (Yang et al, 2020) or via its hinge region that is connected to the DBD (Oksuz et al, 2023; Steiner et al, 2022; Xu et al, 2021). The hinge region of ERα harbors RNA-interacting Arginine Rich Motif (ARM). The mutation in ARM (R269C) causes the loss of RNA binding ability and exhibits the downregulation of Tat-mediated trans-activation of a reporter gene (Oksuz et al, 2023). Conversely, the mutation in RRGG (259–262 aa) within the same ARM exhibits no effect on the binding of ERα to chromatin and transcription in complete media (Xu et al, 2021). The non-reliance of ERα on RNA could be due to the lack of ligand stimulation in this study, as the ligand modulates the binding of ERα on DNA and facilitates the recruitment of robust transcription machinery for target gene activation (Shang et al, 2000). Therefore, the role of RNA in ligand-dependent gene regulation by ERα warrants further functional study.

Toward this, we utilized estrogen stimulation to examine the nuclear role of ERα:RNA interaction. We performed fRIP-seq, biochemical fractionations, microscopy, and genome-wide studies on WT and RBM mutant (RRGG mutant) of ERα in MCF-7 cells upon estrogen stimulation. Through fRIP-seq analysis, we identified interactions of ERα with RNA that transcribe from diverse genomic regions. Further, we observed that the RNA binding-deficient mutant of ERα exhibits poor binding genome-wide upon ligand stimulation. The loss of binding was most predominant on weaker EREs. Additionally, RNase A treatment followed by extraction of chromatin with and without the presence of soluble fraction confirms the requirement of RNA for the stabilization of ERα on low-affinity binding sites. We observed a similar dependence of weaker motifs on RNA for KLF4 and SOX2 binding. Further, FRAP and biochemical

[1]National Center for Biological Sciences, Tata Institute of Fundamental Research, Bangalore, Karnataka 560065, India. [2]SASTRA Deemed University, Thanjavur, Tamil Nadu 613401, India. [3]These authors contributed equally: Bharath Saravanan, Rajat Mann. ✉E-mail: dnotani@ncbs.res.in

fractionations revealed a dynamic binding of ERα on chromatin in the absence of its interactions with RNA. Unexpectedly, we observed that this dynamic binding of ERα leads to increased RNA polymerase II loading and subsequent transcription of E2-regulated genes. Overall, our results suggest that ERα binding on weaker motifs is RNA dependent, where very stable binding either via DNA motif or RNA negatively regulates the transcription.

# Results

## ERα interacts with RNA

ERα interacts with mRNA under complete media (Joyce et al, 2010; Xu et al, 2021) and to specific lncRNAs (Aiello et al, 2016; Horie et al, 2022; Xue et al, 2016). To identify the RNA that interact with ERα after ligand stimulation, we performed fRIP-seq (Hendrickson et al, 2016) post 1 h of estradiol (E2) treatment in MCF-7 cells. The enriched RNAs were from various types of regions in the genome (Fig. 1A). The highest proportions of RNA that interacted with ERα were introns (~30%) and promoter or promoter-proximal regions (~22%), followed by 3′UTR, exon and intergenic RNA. Log2 fold changes over input exhibited stronger interaction with 3′UTR followed by intergenic (Figs. 1B,C and EV1A). Similar to genic regions, ERα exhibited interaction with eRNAs as shown for TFF1e (Fig. 1D). Since the fRIP-seq does not distinguish the RNA from chromatin or nucleoplasmic fraction, we intersected the ERα fRIP-seq peaks with RNA-seq from chromatin or nucleoplasmic fractions in MCF-7 cells (Ntini et al, 2018). ERα fRIP-seq peaks exhibited no preferential biases towards the RNA from a specific nuclear fraction (Fig. 1E). As expected, the introns and distal intergenic regions were more enriched in chromatin fraction, whereas exons were enriched in nucleoplasmic fraction (Fig. EV1B).

The interactions between ERα and TFF1 eRNA were validated through RNA pulldown employing in vitro transcribed TFF1 eRNA (Fig. 1F) and fRIP-PCR (Fig. EV1C). Upon ligand stimulation even though, overall levels of ERα reduced, we observed a strengthening of TFF1 eRNA:ERα interaction (Figs. 1G and EV1D). Similar ligand-dependency was observed for various other RNAs (Fig. EV1E). Since both DNA and RNA bind to ERα, we used unlabeled DNA with the same sequence as RNA, as a competitor in the RNA pulldown assay, which showed RNA:ERα interaction to be specific and unaffected by increasing DNA amount (Fig. 1H). Further, TFF1 eRNA pulldown using biotin-3X ERE (estrogen-response element) oligos as bait was enhanced upon ERα overexpression suggesting ERα can simultaneously bind TFF1 eRNA and the ERE sequence (Fig. 1I). Furthermore, we noted high ERα binding within 10 kb of fRIP-seq peaks (Fig. 1J). Lastly, this trend was more pronounced for the RNA in nucleoplasmic fraction as opposed to the chromatin (Fig. EV1F–H). Together, this data suggests that ERα specifically interacts with a variety of RNAs in a ligand-dependent manner. Further, the binding of ERα in the genome is high near to RNA that interacts with ERα.

## RNA binding mutant of ERα shows loss of binding genome-wide

ERα interacts with RNA through its hinge region (255–272 aa) (Oksuz et al, 2023; Steiner et al, 2022; Xu et al, 2021) where amino acid residues, RRGG (259–262 aa) shows specific binding to RNA. Similar to this report, we created an RNA-binding-deficient mutant of ERα (RBM-ERα) by substituting RRGG to AAAA (Fig. EV2A). The nuclear localization was unaffected (Fig. EV2B). As expected, RBM-ERα in HEK cells that lack endogenous ERα did not show binding with TFF1 eRNA as compared to the WT-ERα (Fig. 2A). Since, ligand strengthens ERα:RNA interactions (Figs. 1G and EV1D; S1E), we hypothesized that lack of RNA binding may affect the chromatin binding of ERα in a ligand-dependent manner. Toward this, we performed chromatin fractionation on cells expressing RBM or WT-ERα upon E2 stimulation and found a considerable reduction of RBM-ERα in chromatin fraction as compared to WT-ERα (Fig. 2B). To extend this observation genome-wide, we performed ChIP-seq of RBM and WT-ERα upon E2 exposure. We noted, decreased binding of RBM-ERα genome wide as compared to the WT-ERα (Figs. 2C and EV2C). Loss of binding was observed for most of the E2-regulated genes, including highly inducible genes like TFF1 and GREB1 (Figs. 2D and EV2D). To rule out the loss of RBM-ERα binding in the genome due to technical reasons, we performed paired-factor ChIP (pfChIP) of ERα and CTCF in the same reaction tube. CTCF was chosen due to its ERα independent binding in the genome (Holding et al, 2018). We plotted the total read counts for CTCF peaks and observed unaffected CTCF binding in WT or RBM-ERα (Fig. 2E). However, RBM-ERα binding was notably reduced (Fig. 2F). These results confirmed the specificity of the reduced genomic occupancy of RBM-ERα. Notably, the loss of RBM-ERα binding was proportional to the level of RNA being transcribed at a given genomic site (Fig. 2G). Furthermore, to rule out the possibility that reduction in ERα occupancy upon RBM expression is not due to the presence of endogenous ERα in MCF-7 cells, we removed endogenous ERα by shRNA against 3′UTR of the ESR1 mRNA (Fig. EV2E). Similar to the previous result, in our ChIP-qPCR (Appendix Table S5), we observed loss of RBM-ERα over WT on GREB1 enhancer, its promoter, and on NRIP1 enhancer (Fig. EV2F)

Additionally, the regions that were bound with ERα and also showed enrichment in fRIP-seq (Fig. EV2G), exhibited more loss of ERα binding upon RBM-ERα mutation comparing regions that do not show RNA binding (Fig. 2H). WT-ERα preferentially binds to intronic and intergenic regions, whereas, RBM-ERα exhibited enrichment on promoters (Fig. 2I). To get better insight into this, we tested the loss of RBM-ERα on promoters, intronic, and intergenic regions and observed that intronic and intergenic peaks lose more ERα upon RBM expression compared to promoters (Fig. 2J). These results align with our fRIP-seq, suggesting the role of RNA in ERα binding at intronic and intergenic regions, whereas promoters are less dependent on RNA.

## Weaker motifs exhibit the highest loss of ERα upon RNA binding deficiency

The lost peaks in case of RBM-ERα were majorly intergenic and intronic while more than 50% of retained sites were promoters (Fig. EV3A). We also observed that ERα:RNA interaction was more at the lost sites as compared to the retained sites (Fig. EV3B). Notably, ERE motif analysis revealed lower scores in lost sites compared to retained ones (Fig. EV3C) suggesting, lost sites were able to bind with ERα due to RNA, despite their weak DNA binding potential. This cumulative contribution of weaker motifs and

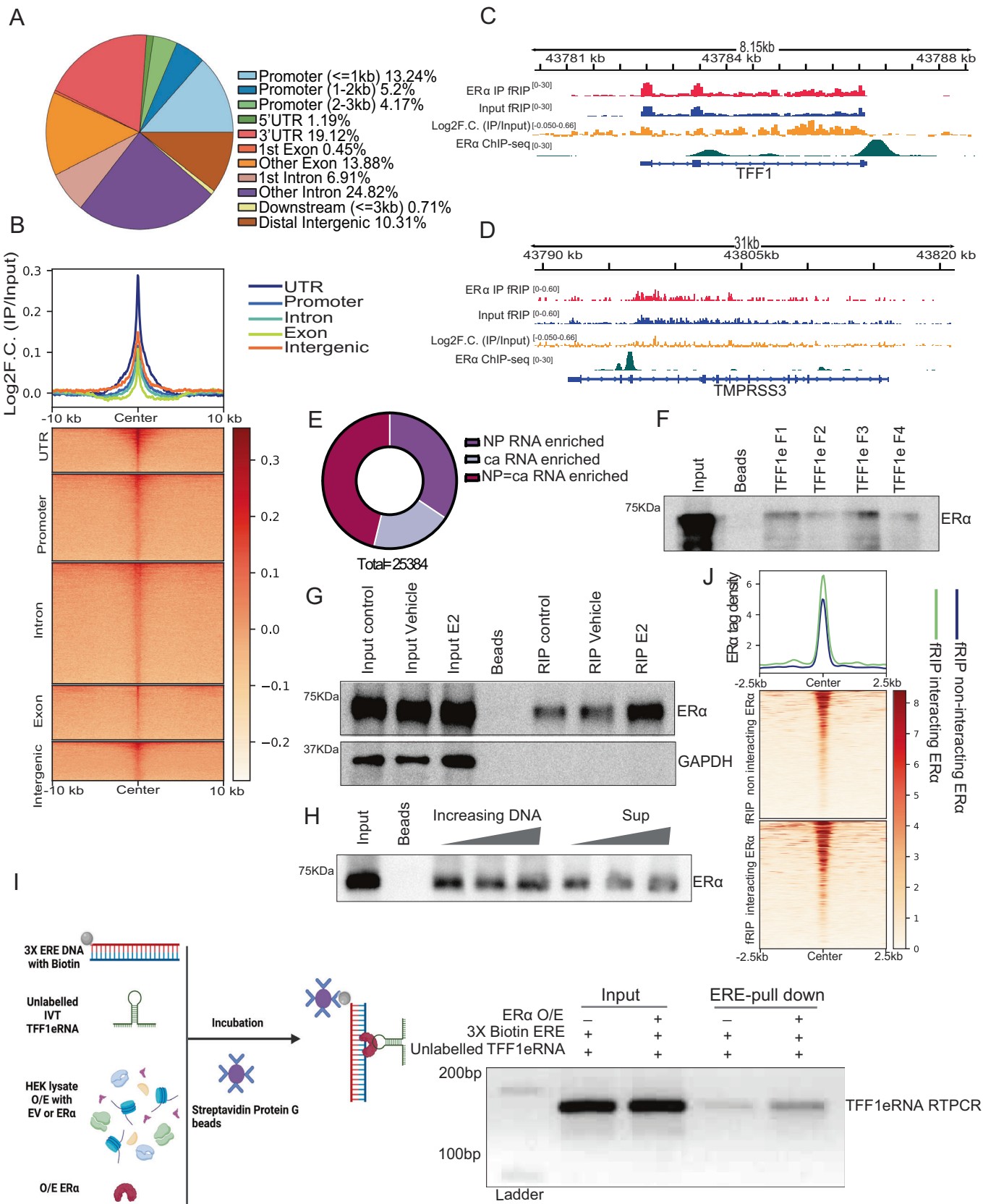

**Figure 1.   ERα interacts with RNA.**

(A) Pie chart showing the genomic distribution of ERα interacting RNA by formaldehyde-assisted RNA immunoprecipitation sequencing (fRIP-seq). (B) Heatmap showing the enrichment of fRIP-seq immunoprecipitation over the input across categories of ERα interacting RNA. (C) Genome browser screenshot showing fRIP-seq IP, Input, Log2F.C.(IP/Input), and ERα ChIP-seq for TFF1 locus. (D) Genome browser screenshot with fRIP-seq IP, Input, Log2F.C.(IP/Input), and ERα ChIP-seq for TFF1 enhancer. (E) Proportions of ERα interacting RNA inside the nucleus (NP nucleoplasmic enriched RNA and ca: Chromatin-associated RNA). (F) Immunoblot for ERα on RNA pulldowns using biotin-labeled TFF1 eRNA fragments. (G) Immunoblot for ERα and GAPDH on RNA pulldowns using biotin-labeled TFF1 eRNA with lysates from cells grown in DMEM or stripping media treated with either Vehicle or E2. (H) Immunoblot for ERα on RNA pulldowns using TFF1 eRNA with increasing concentration of the TFF1 enhancer DNA as a competitor. (I) TFF1 eRNA RTPCR from biotin-labeled 3X ERE as bait with lysates from HEK-293T expressing either empty vector or ERα. (J) Heatmap depicting the strength of ERα binding in intergenic regions within (fRIP interacting sites) and beyond (fRIP non-interacting sites) 10 Kb of fRIP-seq peak. Source data are available online for this figure.

stronger RNA interaction facilitated ERα binding on weaker motifs. Consequently, upon RBM expression, these weaker motifs exhibited higher susceptibility compared to regions with stronger ERE scores.

In order to better understand the relationship between ERE strength and ERα binding, we binned all ERα peaks based on the strength of EREs in descending order. As expected, higher motif scores correlated with higher ERα binding (Fig. 3A). This binding of ERα on stronger EREs was motif driven as they profoundly lost ERα upon mutation in p-box of ERα that perturbs its DNA binding ability (Fig. 3A,B). Contrastingly, the lower strength ERE's were dependent on RNA as these sites lost the most ERα, upon mutation in RBM but were least affected upon DBD perturbations (Fig. 3A,B). Similar observations were made using publicly available data, when RNA was depleted using RNA PolII inhibitor in T47D cells (Zhang et al, 2021) where ERα loss was profound on weaker EREs as compared with stronger EREs (Fig. EV3E). Similarly, utilizing publicly available data for other transcription factors, namely KLF4 and SOX2 in mESCs, a significant loss of these factors upon RBD mutation was observed at lower motif strength (Fig. 3C,D). We validated RNA-dependent recruitment of ERα on TFF1 enhancer by removing TFF1 eRNA using shRNA (Figs. 3E and  EV3F). Together, these results suggest that ERα is recruited on weaker ERE's by RNA.

## Retention of ERα on weaker binding sites depends on RNA

We asked if high RNA-transcribing sites retain more ERα due to the RNA per se. To this end, we focused on non-genic ERα-bound sites from ChIP-seq because ERα exhibits the highest binding on these sites (Li et al, 2013). We grouped these sites into bins of nascent RNA tag counts transcribing from these sites in increasing order and observed a positive correlation between ERα binding and the level of transcription (Fig. 4A). Importantly, this increase in ERα binding was independent of disparities in ERE strength, indicating that under equal ERE strength conditions, high RNA supports high ERα binding (Fig. 4B). Similar positive correlation between RNA and ERα was observed across all ERα-bound sites in the genome (Fig. EV4A,B).

We sought to understand if RNA per se is needed to maintain ERα on chromatin. Toward that, we determined if ERα retention on DNA is dependent on RNA, we performed biochemical chromatin fractionation in the presence of RNAse A and found the loss of ERα in chromatin fraction (Fig. 4C). Similar loss of ERα was also observed on EREs upon removal of total RNA (Fig. 4D). To extend these observations genome-wide, we performed RNase A treatment

on non-crosslinked cells since crosslinking can compensate for the loss of binding (Thakur et al, 2019). We utilized a pre-extraction protocol following RNase A treatment to remove soluble and weakly bound chromatin fractions (Fig. 4E). The presence of this weakly bound subpopulation can yield contrasting results, as previously shown for CTCF (Gu et al, 2020; Saldaña-Meyer et al, 2019) and PRC2 (Beltran et al, 2016; Long et al, 2020). Upon performing ChIP-Seq after RNase A treatment and pre-extraction, we observed a severe loss of ERα binding in the genome (Figs. 4F, H, J and EV4C). Additionally, the loss of binding with pre-extraction suggests a differential reduction, wherein sites transcribing higher RNA showed a more pronounced ERα loss compared to those transcribing less (Fig. 4L). Subsequently, we facilitated the re-binding of ERα by retaining the soluble fraction after RNAse A treatment, followed by crosslinking. This approach led to an excessive binding of ERα in the genome (Figs. 4G, I, K and EV4C). This increased ERα binding in the genome was driven purely based on the strength of the ERE because RNA was already degraded by RNase A (Fig. 4M). The nuclear levels of ERα remained unaffected with and without RNASe A treatment (Fig. EV4D). Collectively, these results suggest that RNA contributes to the stabilization of ERα on weaker ERE motifs, and in the absence of RNA, ERα interacts weakly with the weaker EREs (Fig. EV4E).

## RNA binding mutant of ERα interacts dynamically with the chromatin

The ChIP-seq results showed the reduced binding of RBM-ERα on its genomic sites (Figs. 2C and EV2C). Additionally, the loss of ERα on chromatin after removal of total RNA suggests that RNA regulates the strength of ERα interaction with the chromatin (4C, D, F, H). We hypothesized that the lack of RNA binding in the case of RBM-ERα could be associated with its dynamic molecular interactions with chromatin. To test this, we performed FRAP on WT:GFP and RBM-ERα:GFP in live cells (Fig. 5A). We observed that recovery of RBM-ERα after photobleaching was much faster with median t ½ of 8 s as opposed to 12 s in the case of WT-ERα (Fig. 5B,C). ERα forms condensate that are linked to transcription (Saravanan et al, 2020, Nair et al, 2019). We asked if the increased mobility would affect the condensates of RBM-ERα:GFP. Indeed, RBM-ERα condensates exhibited a significantly reduced normalized area fraction occupied than WT (Fig. 5D). The dynamic binding seen in FRAP, could be due to its low-affinity interactions with the chromatin; we tested affinity of RBM-ERα to chromatin by retention assay where weak chromatin-associated proteins are removed from the nucleus using CSK (cytoskeleton) buffer. We noted loss of RBM-ERα over the WT-ERα confirming its weaker chromatin

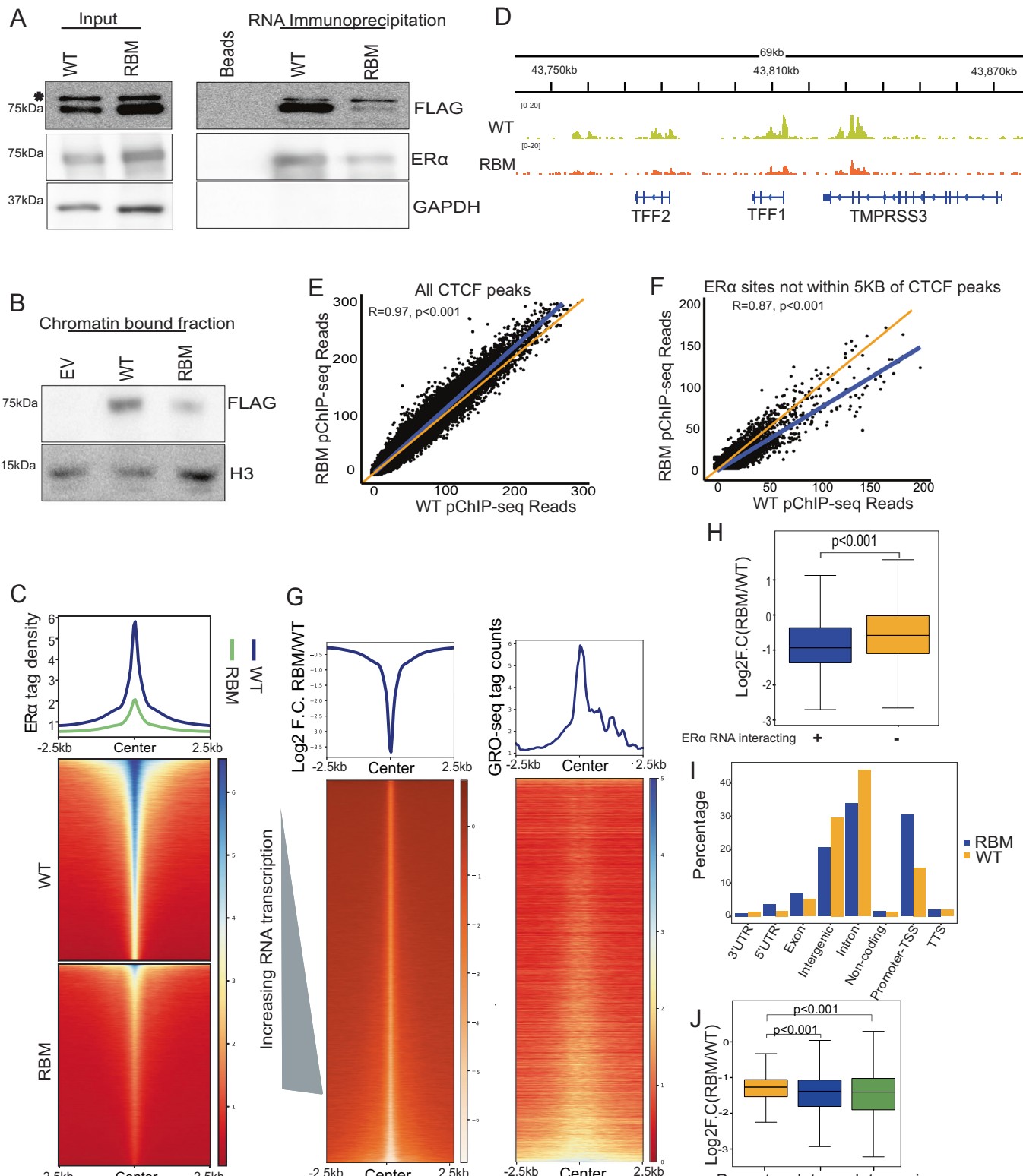

interactions (Fig. 5E). Further, we confirmed the low-affinity interactions of RBM-ERα on chromatin by fractionation of ERα from chromatin at varying salt concentrations. Indeed, the binding of the mutant was of low affinity as seen by its elution from chromatin at low salt concentration (75–150 mM) as compared to the WT-ERα

(300 mM) (Fig. 5F). Similar results were obtained upon the removal of total RNA with RNAse A (Fig. 5G). These results suggest that RNA binding is required for stable binding of ERα to the chromatin as the loss of RNA or RNA-binding results in more dynamic interaction of ERα with chromatin and higher nuclear mobility.

Figure 2.   RNA binding mutant of ERα shows loss of binding genome-wide.

(A) Immunoblot for FLAG, ERα and GAPDH on RNA pulldowns using TFF1 eRNA for ERα WT and RBM overexpressed lysates from HEK-293T (* denotes non-specific band). (B) Immunoblot for FLAG and H3 on chromatin-bound fraction from ERα WT or RBM overexpressed MCF-7 cells. (C) Heatmap depicting the binding strength of ERα:FLAG WT and RBM overexpressed in MCF-7 cells. (D) Genome browser screenshot for the binding of ERα WT and RBM on TFF1 locus in MCF-7. (E, F) Normalized read count of ERα WT and RBM pfChIP-seq for all CTCF peaks and ERα peaks beyond 5 kb of CTCF peaks, respectively. Here r denotes Pearson correlation and p value is calculated using t-test. (G) Heatmap depicting the log2F.C. of binding of RBM-ERα over WT-ERα and GRO-seq signal plotted at the sorted sites in the same order. (H) Boxplot depicting the Log2F.C. (RBM/WT) at 2781 ERα sites categorized as either RNA-interacting or non-interacting. Statistical significance determined by Mann–Whitney U-test. (I) Genomic distribution in percentages for the ERα:FLAG WT and RBM. (J) Boxplot depicting the Log2F.C. (RBM/WT) at ERα sites categorized as promoters (within 500 bp of 14,206 ERα peaks), intronic (50,302 peaks), and intergenic (26,992 peaks). Statistical significance was determined using the Mann–Whitney U-test. The center lines of the boxplot denote the median, the box limits indicate the 25th and 75th percentiles, and the whiskers extend 1.5 times the interquartile range from the 25th and 75th percentiles. Outliers are not presented. Replicates for the WT and RBM ChIP-seq biological replicates in C, D and H–J is provided in Appendix Table S6. Source data are available online for this figure.

## The dynamic binding of ERα allows better transcription of target genes

To understand the impact of dynamic RBM-ERα binding on gene transcription, we conducted reporter assays using 3X ERE upstream of the luciferase cassette in MCF-7. The RBM-ERα exhibited significantly higher luciferase activity compared to the WT-ERα (Figs. 6A and EV5A). To eliminate the possibility of these effects arising due to endogenous ERα, we performed reporter assays where we reduced endogenous ERα levels by shRNAs (Fig. EV5B) and overexpressed WT and RBM-ERα in this background. The results confirmed the heightened transcriptional potential of RBM-ERα (Fig. 6B). We extended the reporter assays in HEK cells that don't express ERα and found similar results (Fig. EV5C). Expanding on this finding, we employed nascent EU-seq and observed a set of upregulated genes in cells expressing RBM-ERα over WT-ERα in MCF-7 (Fig. 6C). Furthermore, we noticed a higher enrichment of PolII-S2p in the chromatin fraction of cells expressing RBM-ERα (Fig. 6D). To corroborate these observations, we performed total PolII ChIP-seq which also confirmed the greater occupancy of total PolII on the promoters of upregulated genes identified from EU-seq (Figs. 6E,F and EV5D). The increased PolII enrichment in the case of RBM-ERα was further confirmed by Drosophila spike-in DNA (Fig. EV5E). RBM-ERα had a more pronounced effect on the highly E2-induced genes (referred to as "changing" genes), as compared to less robustly E2-induced genes (referred to as "Non-changing") (Fig. 6G, H). The changing genes also exhibited higher PolII occupancy compared to the non-changing and random genes upon RBM-ERα expression (Fig. 6I). The extent of PolII gain on the gene body of these changing genes was similar to the gain observed upon E2 treatment compared to vehicle (Fig. 6J).

Additionally, we observed a significant portion of ERα binding to intronic regions and intronic RNA by ERα ChIP-seq and fRIP-seq, respectively (Figs. EV5F and 1A). Furthermore, we observed intergenic and intronic regions to harbor weaker EREs as compared to promoters. (Fig. EV5G,H). Notably, intronic enhancers associated with changing genes exhibited higher ERα occupancy compared to intronic enhancers from the non-changing category (Fig. EV5I). The most upregulated genes upon E2 induction, such as TFF1 and GREB1, are regulated by intergenic and intronic megatrans enhancers (Liu et al, 2014). We observed these two genes in changing gene categories. We attributed this heightened enrichment of ERα at enhancers of changing genes to the increased interaction of intronic RNA with ERα (Fig. EV5J). This suggests that RNA:ERα interaction contributes to ERα stabilization on

highly active intronic/intergenic enhancers (Fig. 6L) of genes that are upregulated upon RBM-ERα expression.

Consequently, upon RBM-ERα expression, ERα peaks from introns of changing gene category exhibited a more pronounced loss of ERα compared to non-changing genes, indicative of a transition to dynamic binding associated with enhanced transcription (Fig. 6K). To validate such modulation of ERα-mediated transcriptional response, we conducted in vitro luciferase reporter assays with increasing concentrations of RNA. The results indicate that at lower RNA concentrations, there is an enhancement in transcription. However, at higher RNA concentrations, mimicking situations where RNA accumulates due to transcription, there is an inhibition of transcription (Fig. EV5K). The lack of RNA binding, RBM-ERα, allows it to bind on target sites in a dynamic manner, which favors the robust PolII recruitment and transcription (Fig. 7).

## Discussion

### ERα interacts with RNA

The interaction of TFs with DNA was assumed to be the canonical mode of TF binding on chromatin. Recent studies dissecting the functional role of TF-RNA interactions have revealed that RNA modulates the TF's interaction with chromatin leading to transcription regulation. The interaction between ERα and RNA has been shown (Oksuz et al, 2023; Steiner et al, 2022; Xu et al, 2021; Yang et al, 2020). However, its role in ERα binding on chromatin and downstream effects on transcription is debated from presumed no effect on DNA binding and transcription under complete media (Xu et al, 2021) to a significant effect on transcriptional activation (Oksuz et al, 2023). The ligand modulates the binding of ERα on DNA and facilitates the recruitment of robust transcription machinery for target gene activation (Shang et al, 2000), suggesting ERα:RNA interaction and its function may be influenced by the ligand-dependent events. Therefore, we utilized E2 signaling to examine the nuclear role of ERα:RNA interaction as opposed to under complete media (Xu et al, 2021). Our findings reveal that ERα binds to various coding and non-coding RNAs (Fig. 1A), and this interaction is enhanced upon E2 stimulation (Figs. 1G and EV1D,E). Specifically, we confirm that ERα interacts with RNA through the hinge region in the C-terminal extension (CTE) of its DNA-binding domain (DBD) (Figs. 2A and 1H,I). This RNA-dependent mechanism of ERα recruitment differs from previous findings (Yang et al, 2020) where repressive eRNAs

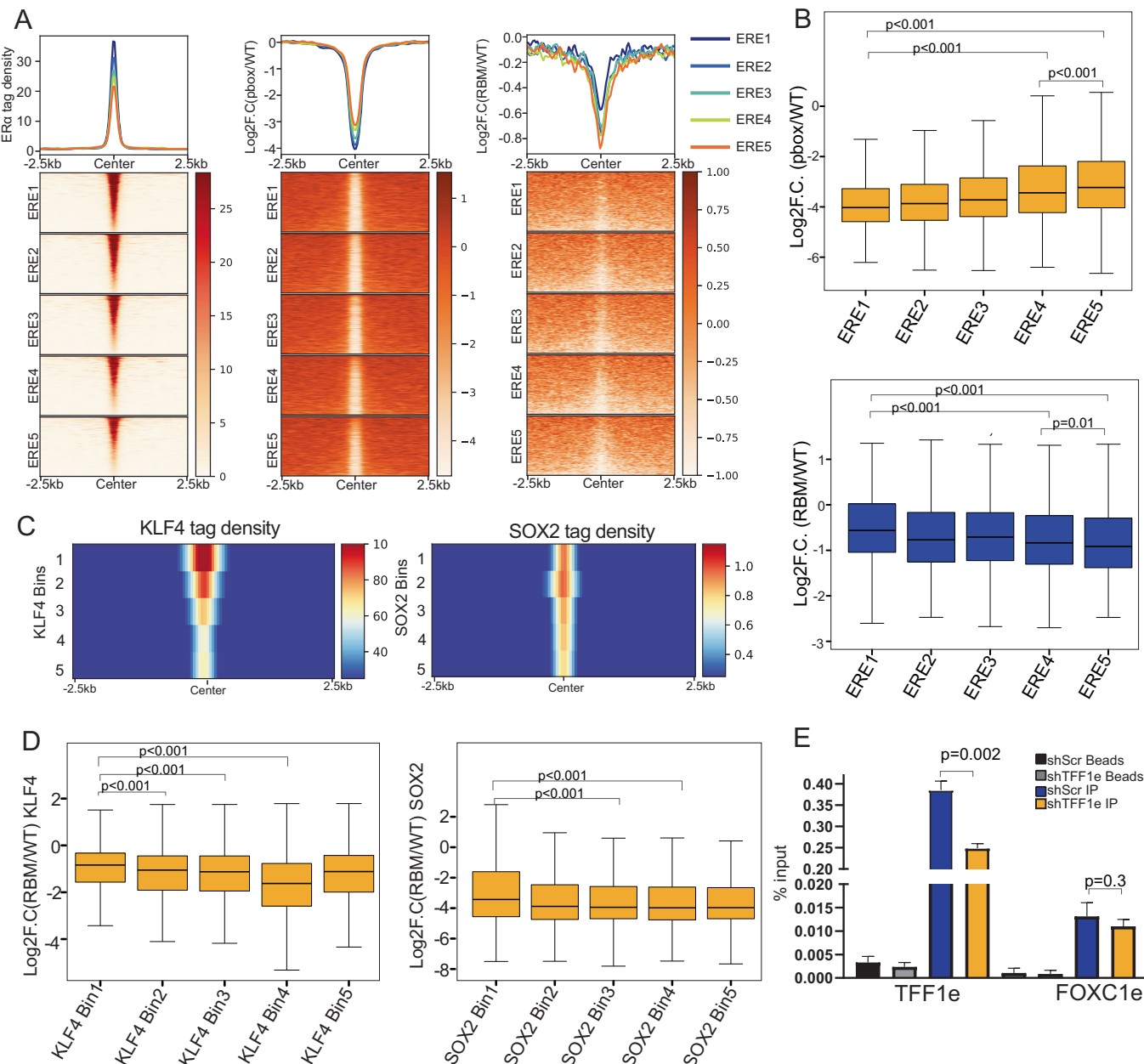

**Figure 3. Weaker motifs exhibit the highest loss of ERα upon RNA binding deficiency.**

(A) Heatmap depicting the ERα tag density, Log2F.C (pbox/WT) and (RBM/WT) at ERE bins of varying strength (strongest to weakest). (B) Boxplot depicting the Log2F.C. (pbox/WT) and (RBM/WT) at decreasing order of ERE strength bins each consisting of 1060 sites. Statistical significance determined by Mann–Whitney *U*-test. (C) Profile plot illustrating the KLF4 and SOX2 tag density at their respective motif bins of varying strength (strongest to weakest). (D) Boxplot depicting the Log2F.C. (RBM/WT) for KLF4 and SOX2 at decreasing order of motif strength, with each bin consisting of ~9000 sites for KLF4 and ~6500 sites for SOX2. Statistical significance determined by Mann–Whitney *U*-test. (E) ERα enrichment on enhancers of TFF1 and FOXC1 upon transfecting short hairpin RNA targeting either scramble or TFF1 enhancer RNA (sense and antisense). Statistical significance was determined by unpaired *t*-test, and error bars denote the standard error of the mean (SEM) from three biological replicates. The center lines of the boxplot denote the median, the box limits indicate the 25th and 75th percentiles, and the whiskers extend 1.5 times the interquartile range from the 25th and 75th percentiles. Outliers are not presented. Replicates for (A, B), WT and RBM-ERα ChIP-seq and for (C, D) publicly available datasets for KLF4 and SOX2 WT and RBM CUT&Tag are mentioned in Appendix Table S6. Source data are available online for this figure.

interacted with ERα solely through the DBD thus, precluding simultaneous interactions with both DNA and RNA. Our findings align with earlier studies on GR and Sox2, demonstrating that these transcription factors can interact with specific RNA either via their DBD or the CTE of DBD (Holmes et al, 2020; Hou et al, 2020;

Lammer et al, 2023; Parsonnet et al, 2019), thereby modulating their DNA occupancy. We provide evidence supporting the functional implications of ERα's concurrent DNA-RNA binding whereby, RNA interactions with ERα result in higher occupancy compared to non-interacting sites (Fig. 1J). The binding of ERα to

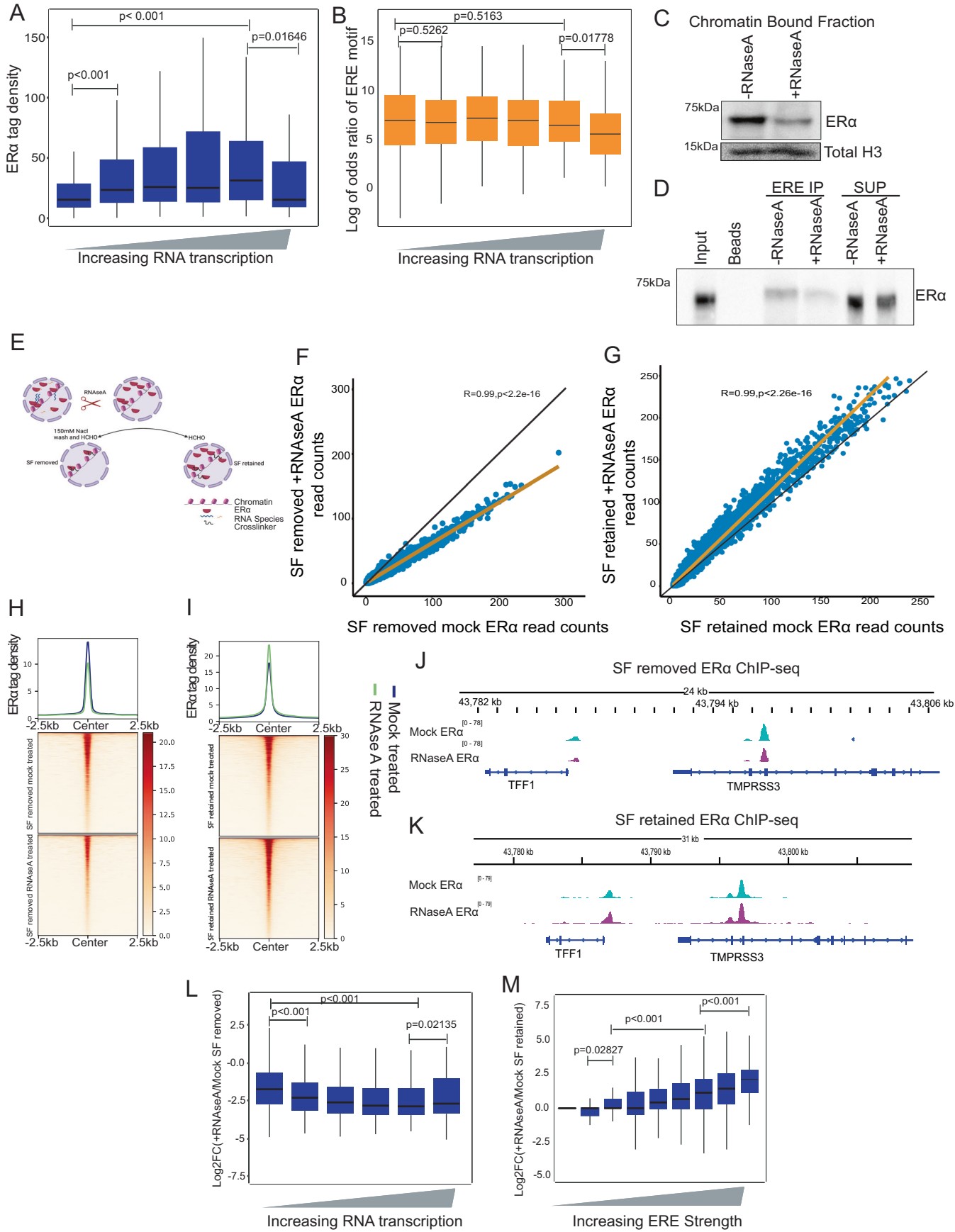

**Figure 4. Retention of ERα on weaker binding sites depends on RNA.**

(A) Boxplot showing ERα enrichment on all non-genic sites binned based on the levels of RNA transcription in increasing order. (B) Boxplot showing the Log of odds ratio for the ERE motif on all non-genic sites binned on the basis of RNA transcription in increasing order. (C) Immunoblot of ERα and total H3 on chromatin-bound fractions from nucleus with or without RNase A treatment. (D) ERE binding to ERα in cellular lysate in the presence or absence of total RNA. (E) Schematic of ChIP-seq following the removal of total RNA before crosslinking, with and without pre-extraction. (F, G) Scatter plots showing a correlation between ERα ChIP-seq normalized read counts with and without RNase A following pre-extraction and in the absence of pre-extraction respectively. Here r denotes Pearson correlation and p value is calculated using t-test. (H, I) Heatmap depicting ERα tag density between mock and RNase A treatment followed by pre-extraction and without pre-extraction, respectively. (J, K) Genome browser screenshot of TFF1 locus showing ERα ChIP-seq upon RNase A treatment with pre-extraction of soluble proteins and with retention of soluble proteins, respectively. (L) Log2FC of +RNase A/mock with pre-extraction ERα ChIP-seq at all sites binned based on the RNA transcription. Statistical significance determined by Mann–Whitney U-test. (M) Log2FC of +RNase A/mock SF retained at all sites binned based on increasing ERE strength. Statistical significance determined by Mann–Whitney U-test. The center lines of the boxplot denote the median, the box limits indicate the 25th and 75th percentiles, and the whiskers extend 1.5 times the interquartile range from the 25th and 75th percentiles. Outliers are not presented. In (A, B, L), the first to sixth bins contain 9104, 438, 572, 432, 173, and 102 regions, respectively. In (M), the first to sixth bins contain 2698, 4554, 1880, 750, 394, and 545 peaks, respectively. Replicates for (A, B), publicly available ERα ChIP-seq and GRO-seq. For (F–M) Mock and RNAse A-treated ERα ChIP-seq are mentioned in Appendix Table S6. Source data are available online for this figure.

RNA offers an additional mechanism for its chromatin tethering while its DBD engages with ERE on DNA.

## RNA stabilizes ERα binding on weaker ERE motifs

Owing to ERα's capability to bind RNA, our data suggest that transcribed sites with similar ERE strength can recruit more ERα as compared to non-transcribing sites (Fig. 4A,B). Furthermore, pronounced loss of RBM-ERα on weak EREs, as opposed to the loss of DBD mutant ERα on strong EREs (Figs. 3A,B and EV3A–C) confirms the RNA mediated binding of ERα on weaker EREs.

In addition to the major groove interactions mediated by the DBD, the CTE of nuclear receptors establishes additional contact with the minor groove of DNA (Jakób et al, 2007; Gearhart et al, 2005; Zhao et al, 1998). Based on these facts, we propose that the interaction of RNA with the CTE of ERα, can stabilize ERα on DNA. Moreover, we demonstrate that this RNA-dependent stabilization of transcription factors on weaker motifs (Fig. 3C,D) also applies to other transcription factors such as KLF4 and Sox2. Additionally, it has been shown that CTCF-RBD mutants exhibit reduced occupancy on low-affinity sites, which predominantly occur at promoters (Saldaña-Meyer et al, 2019). Furthermore, in the absence of RNA or upon RBM mutation, ERα loses its ability to contact RNA that leads to the destabilization of ERα on chromatin and results in its dynamic binding behavior. These observations were supported by retention assays, salt elution, and FRAP (Figs. 4 and 5). This increased mobility may arise from the loss of RNA-mediated entrapment, as RNA can act as a crosslinker. The absence of this crosslinking effect may allow ERα to be more available for chromatin binding in a dynamic manner. Indeed, the inability of TFs to interact with RNA increases its unbound fraction in the nucleus, as revealed by single particle tracking (Oksuz et al, 2023).

## Dynamic binding of TF with chromatin correlates with robust transcription

As a result of polymerase loading, the transcription takes place in bursts as opposed to constant transcription (Rodriguez et al, 2019; Pomp et al, 2024). The occurrence of bursts is anti-correlated with the binding of transcription factors (Charoensawan, Martinho, and Wigge 2015; Doidy et al, 2016; Para et al, 2014; Pownall et al, 2023; Schaffner 1988). Therefore, the high burst frequencies would require not the static binding of TF but a dynamic binding where each binding event needs to be long enough (dwell time) to result in subsequent transcription bursts. Indeed, it is now established that

TF binding is much more dynamic as opposed to long prevailed static binding model (Hager et al, 2009; Hansen et al, 2019; Swinstead et al, 2016), and further, stable binding is inhibitory to transcription (Guan et al, 2019; Haelens et al, 2007; de Jonge et al, 2020).

In this direction, soon after ligand stimulation, the basal eRNA at poised enhancers may potentially assist in ERα trapping despite weaker EREs. However, as the eRNA/RNA starts to accumulate, the high RNA may stabilize ERα to the extent that it becomes inhibitory to sustain transcription. In this regard, enhanced mobility of ERα-RBM potentially results in higher PolII occupancy and transcription of E2-regulated genes (Figs. 5A–C and 6). Each dynamic binding event of mutant ERα might be long enough to engage in PolII loading. Further, these observations are consistent with previous reports showing that the dynamic binding of ERα and AR to chromatin leads to its enhanced transcriptional activity (Guan et al, 2019; Kim et al, 2022). Similarly, in the case of AR, mutations in the hinge region have been reported to cause increased transcription and mobility inside the nucleus (Buchanan et al, 2001; Haelens et al, 2007; Tanner et al, 2010). Overall, interaction with RNA spatially confines ERα that negatively impacts the transcriptional upregulation of a subset of E2-regulated genes.

## ERα:RNA interaction under basal signaling vs post-ligand stimulation

Under basal signaling, unliganded ERα poorly binds to chromatin and transcription-activating machinery (Shang et al, 2000) and, in complete media, ERα preferentially interacts with 3′UTR of mRNAs to regulate the integrated stress response through their translation regulation (Xu et al, 2021). Understandably, in complete media, Xu et al, did not observe the loss of RBM-ERα binding on the genome and, therefore, the effect on transcription. Similar to Xu et al, our study also focuses on the RRGG motif in the hinge domain of ERα but under ligand stimulation.

Similar to our and Xu et al, 2021, Oksuz et al, 2023 have also revealed the interaction of ERα with RNA through its arginine-rich motif (ARM) specifically, at R269 in the same hinge domain (255–272 aa). Further, similar to our observation, the RBM of ERα (R269C) also shows a loss of binding to the RNA. However, Oksuz et al show reduced trans-activation potential of ARM-ERα (R269C) using Tat-reporter assays. This assay was performed by using ERα-ARM and mutant ERα-ARMα (R269C) peptide truncation without

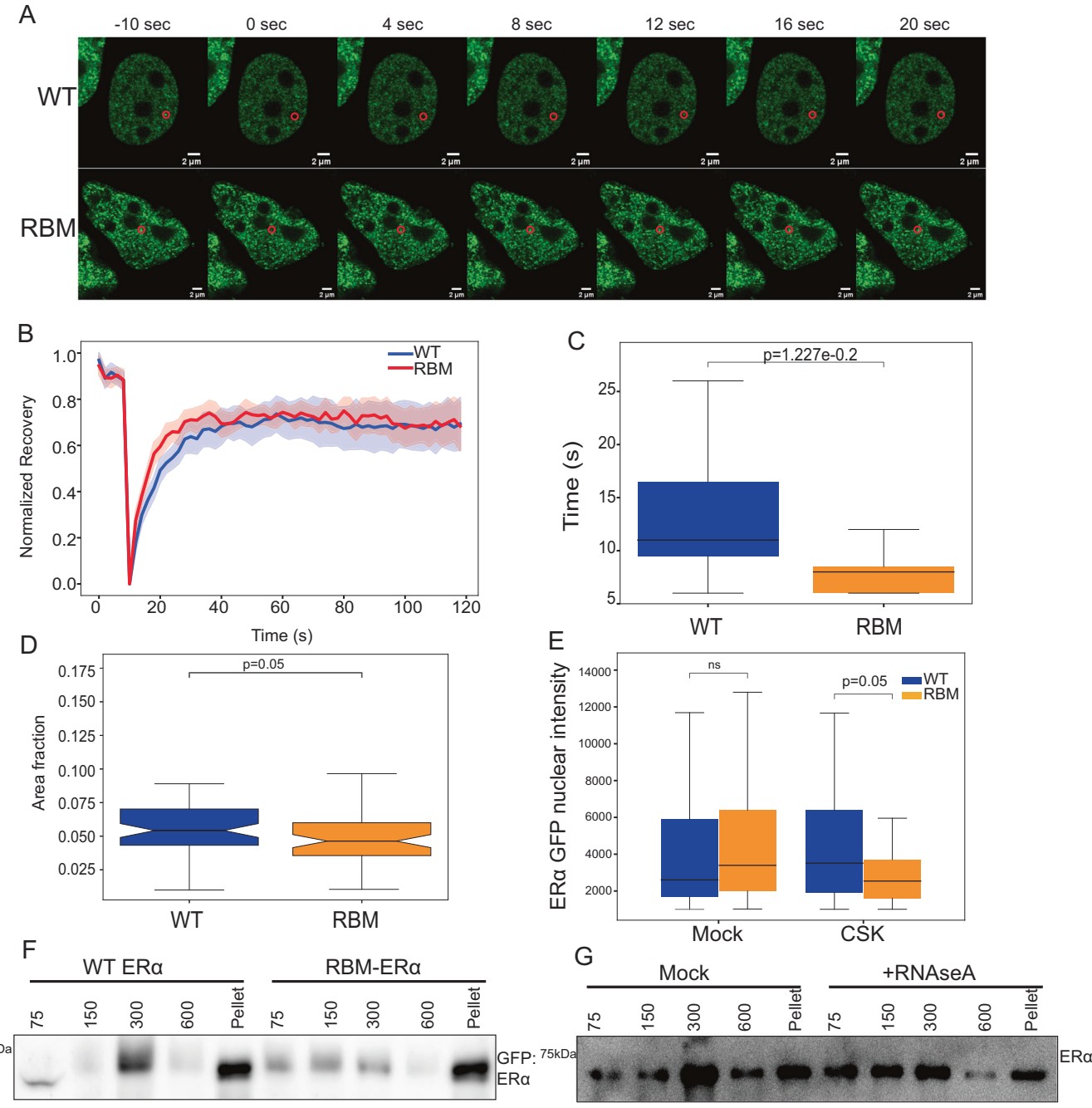

**Figure 5. RNA binding mutant of ERα interacts dynamically with the chromatin.**

(A) Representative image of WT-ERα:GFP and RBM-ERα:GFP showing the FRAP ROI pre and post-bleaching. Red circles denote the ROI that was bleached and followed post-recovery. (B) Recovery plot of FRAP ROIs for WT-ERα:GFP and RBM-ERα:GFP. (C) Boxplot depicting the half-life recovery for WT-ERα:GFP and RBM-ERα:GFP. Statistical significance determined by unpaired *t*-test and error bar denotes SEM calculated from three biological replicates of live cell microscopy, with 15 ROIs (Region of Interest) each for WT and RBM-ERα GFP. (D) Boxplot depicting the normalized area fraction of WT and RBM-ERα. Statistical significance determined by Mann–Whitney *U*-test and error bar denotes SEM calculated from three biological replicates of microscopy with 60 nuclei per condition for WT and RBM-ERα:GFP. (E) Boxplot depicting the retention assay for WT-ERα:GFP and RBM-ERα:GFP. Statistical significance determined by Mann–Whitney *U*-test and error bar denotes SEM calculated from three biological replicates of microscopy with 64, 57, 110, and 79 nuclei for WT mock, RBM mock, WT CSK, and RBM CSK treated, respectively. (F) Immunoblotting for ERα on chromatin-associated proteins eluted at different salt concentrations from cell expressing either WT or RBM ERα:GFP. (G) Immunoblotting for ERα on chromatin-associated proteins isolated from mock and RNAse A-treated nuclei at different salt concentrations. The center lines of the boxplot denote the median, the box limits indicate the 25th and 75th percentiles, and the whiskers extend 1.5 times the interquartile range from the 25th and 75th percentiles. Outliers are not presented. Source data are available online for this figure.

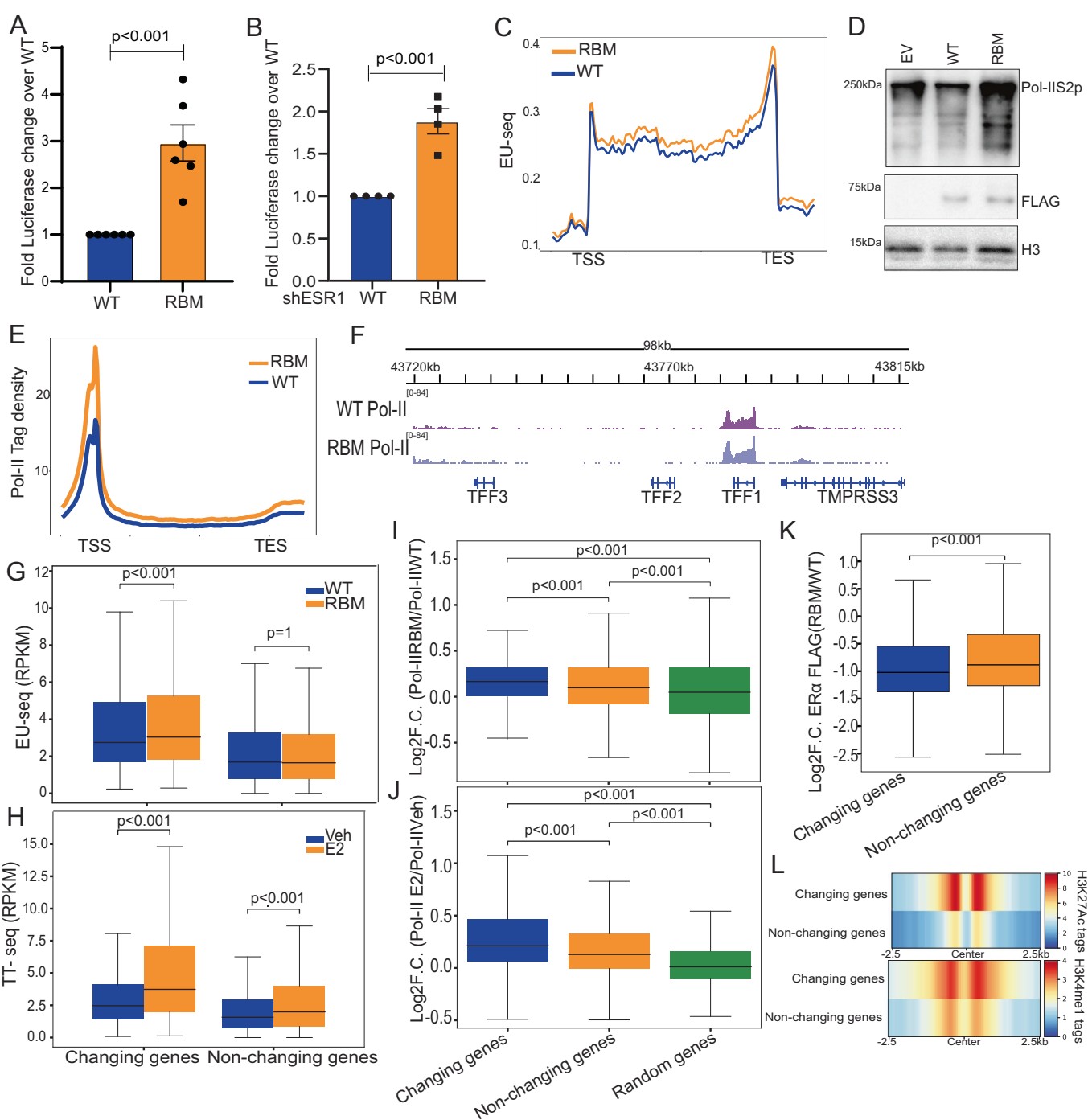

the DNA binding domain of ERα.Tat-reporter relies on the binding of test peptide on RNA to activate the downstream reporter. Therefore, the reporter tested the inability of ERα mutant peptide to bind with RNA and not the effect on the transcriptional potential of mutant ERα via its DNA binding ability. Indeed, similar to our observation, the full length ERα reporter of R269C also show the activation of reporter activity as shown previously (Boldes et al, 2020). Further, the mutation in different hinge residues in ours and Oksuz et al may result in varying transcriptional responses

# Methods

## Cell culture

MCF-7 and HEK-293T cells were obtained from ATCC and maintained in high glucose DMEM (Invitrogen) at 37°C with 5% $CO_2$. For hormone deprivation, MCF-7 cells were seeded in complete DMEM, and the following day, cells were washed with 1X DPBS and then cultured in DMEM without phenol red (Invitrogen) supplemented with 5% charcoal-stripped FBS (Invitrogen). After

**Figure 6.  The dynamic binding of ERα allows better transcription of target genes.**

(A) Luciferase activity normalized to the WT for 3X ERE reporter assay with ERα WT and RBM overexpression in MCF-7 upon 4 h of E2 treatment. Statistical significance determined by Mann–Whitney *U*-test and error bar denotes SEM calculated from six biological replicates. (B) Luciferase activity normalized to WT for 3X ERE reporter assay with ERα WT and RBM overexpression in MCF-7 cells upon 24 h E2 treatment, in the background of endogenous ERα knockdown using shRNA. Statistical significance was determined using the Mann–Whitney *U*-test, and error bar denotes SEM from four biological replicates. (C) Summary plot showing the EU-seq reads from the whole gene body for upregulated genes upon RBM-ERα expression over WT-ERα. (D) Immunoblot of PolIIS2p, FLAG, and H3 from chromatin-bound fractions of either empty vector or WT-ERα or RBM-ERα expressing nuclei. (E) Summary plot showing the total PolII ChIP-seq reads from the whole gene body for upregulated genes upon RBM-ERα expression over WT-ERα. (F) Genome browser screenshot of TFF1 locus for the total Pol-II ChIP-seq upon ERα WT or RBM overexpression. (G) Boxplot showing the EU-seq reads for the whole gene body in two categories- changing genes and non-changing genes in WT-ERα or RBM-ERα. Statistical significance was determined by Wilcoxon signed rank-sum test. (H) Boxplot showing the TT-seq reads for the whole gene body upon vehicle or E2 treatment in two categories- changing genes and non-changing genes. Statistical significance was determined by Wilcoxon signed rank-sum test. (I) Boxplot depicting the Log2F.C. PolII ChIP-seq (RBM-ERα/WT-ERα) for three categories namely, changing genes, non-changing genes, and random gene list. Statistical significance determined by Mann–Whitney *U*-test. (J) Boxplot depicting the Log2F.C.  PolII ChIP-seq (E2/Veh) for three categories namely, changing genes, non-changing genes, and random gene list. Statistical significance determined by Mann–Whitney *U*-test. (K) Boxplot depicting the Log2F.C. of ERα (RBM/WT) on intronic ERα sites from changing (859 peaks) and non-changing (387 peaks) gene categories. Statistical significance determined by Mann–Whitney *U*-test. (L) Profile plot depicting the tag density of H3K27ac and H3K4me1 at intronic ERα sites from changing and non-changing gene categories. The center lines of the boxplot denote the median, the box limits indicate the 25th and 75th percentiles, and the whiskers extend 1.5 times the interquartile range from the 25th and 75th percentiles. Outliers are not presented. For (G–J), the number of regions for changing, non-changing, and random gene categories are 556, 1266, and 1000, respectively Replicates for WT and RBM expression EU-seq (C, G), Pol-II ChIP-seq (E, F, I), and vehicle and E2-treated TT-seq (H), and PolII ChIP-seq (J) are mentioned in Appendix Table S6. Source data are available online for this figure.

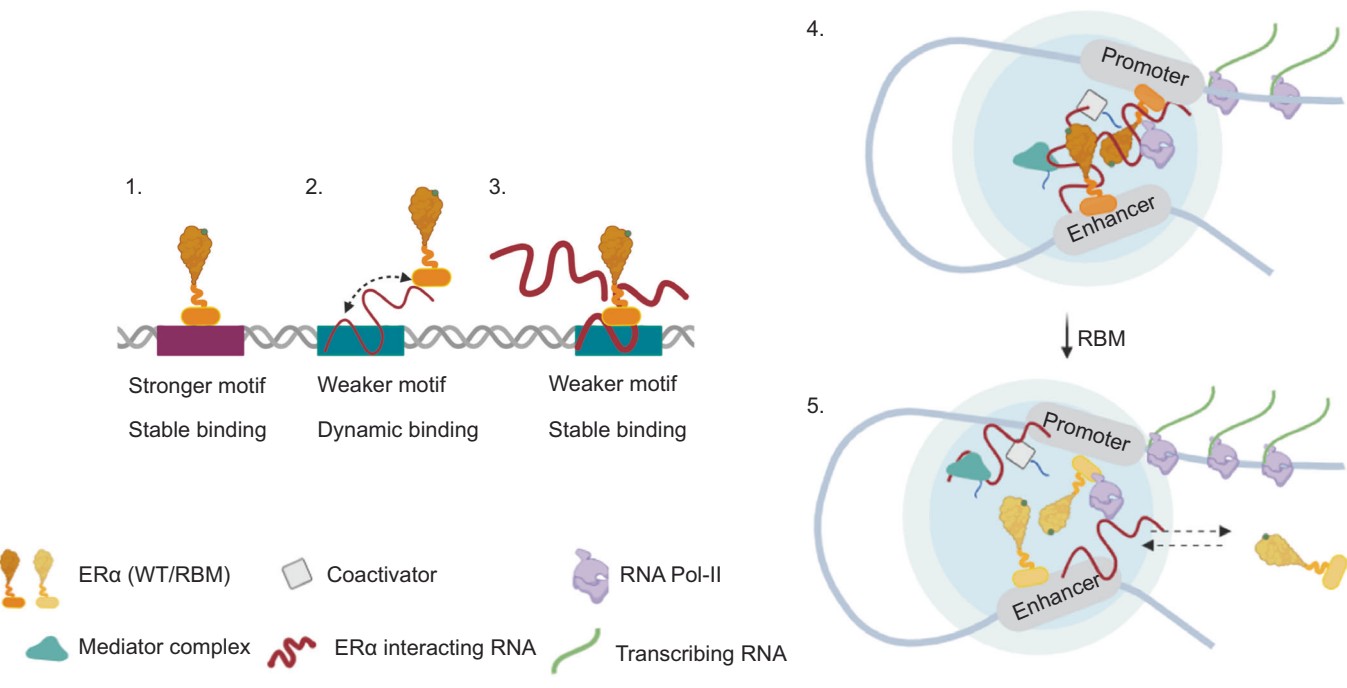

**Figure 7.   Schematic.**

(1) ERα exhibits stable binding to stronger motifs due to high interaction strength between its DBD and DNA. (2) On weaker motifs with low RNA transcription, ERα binds dynamically due to the low-affinity interaction between its DBD and DNA. (3) ERα binding to weaker motifs, coupled with increased RNA accumulation from enhanced signaling-induced transcription, stabilizes the interaction between ERα and DNA through additional interaction of the hinge region of ERα with RNA as positive feedback. (4) The excess stabilization of ERα-DNA interaction by RNA can lead to the formation of ERα condensates, which are less dynamic, inhibiting continuous ligand-induced transcription. (5) Mutation in the RBD of ERα results in dynamic interaction between ERα and DNA, preventing the excess stabilization of ERα in condensates, leading to continuous activation of ligand-induced transcription.

72 h of hormone deprivation, the cells were treated with either 100 nM 17β estradiol (Sigma E2758) or vehicle (ethanol, Millipore) for 1 h for all experiments. For the overexpression studies, transfection was performed using Lipofectamine 2000 (Invitrogen 11668019) according to the manufacturer's protocol in OptiMEM with reduced serum and no phenol red (Invitrogen 11058021) after 48 h of hormone deprivation. After 6 h of transfection, the media

was changed to stripping media. On the third day, which is after 24 h of transfection and a total of 72 h of serum deprivation in cells, the cells were stimulated with the ligand.

For shRNA-mediated knockdown of endogenous ERα, MCF-7 cells were seeded. On the second day of seeding, cells were washed with 1X DPBS cultured in phenol-free media, and transfected using Lipofectamine 2000, either with shRNA targeting the 3′UTR of

ESR1 or a scramble sequence. Six hours after transfection, the media was changed to phenol-free media. After 24 h of transfection (second day of stripping), a second round of transfections was performed with plasmids expressing either WT-ERα or RBM-ERα. Following this, on the third day of stripping (a total of 72 h of serum starvation), cells were stimulated with 17β estradiol and processed for experiments.

## Formaldehyde-assisted RNA immunoprecipitation (fRIP)

The fRIP procedure was performed following the previously described protocol (Hendrickson et al, 2016). Briefly, MCF-7 cells were crosslinked with 0.1% formaldehyde for 10 min at room temperature (RT) with gentle shaking. The crosslinking reaction was quenched by the addition of 0.125 M glycine for 5 min at RT. The cells were then washed, pelleted, and stored at −80 °C. For cell lysis, ~20 million cells were resuspended in 1250 µl of RIPA lysis buffer (50 mM Tris pH 8, 150 mM KCl, 0.1% SDS, 1% Triton X, 5 mM EDTA, 0.5% sodium deoxycholate, 0.5 mM DTT, 1X protease inhibitor cocktail (PIC), 100 U/ml RNase inhibitor) and incubated on ice for 10 min. Then the lysates were sonicated using a Bioruptor Pico (from Diagenode) for six cycles with 30 s ON and 30 s OFF program. The lysates were then clarified by centrifugation at 15,000 rpm for 10 min, and the supernatant was diluted by adding an equal amount of fRIP wash buffer (150 mM KCl, 25 mM Tris pH 7.5, 5 mM EDTA, 0.5% NP-40, 0.5 mM DTT, 1X PIC, 100 U/ml RNase inhibitor). The diluted lysates were passed through a 0.45 µM membrane syringe filter. For immunoprecipitation, 1 µg of anti-ERα antibody (sc-8002) was added to 1 ml of lysate, followed by incubation at 4 °C for 2 h. After incubation, 20 µl of Protein G beads were added, and the mixture was incubated for 1 h on a rotor at 4 °C. The antibody-protein-bead complexes were then washed thrice with fRIP wash buffer at high speed on the rotor.

## For RNA elution, purification, and fRIP-PCR/fRIP-seq

The beads were resuspended in 56 µl of nuclease-free water (NFW). Then, 33 µl of elution buffer was added to each tube. The samples were incubated at 42 °C for 1 h, followed by an additional incubation at 55 °C for 1 h. TRIzol was added to the tube to extract RNA according to the manufacturer's instructions. The purified RNA was used for cDNA synthesis using the SuperScript™ IV First-Strand Synthesis System (#18091050) or for preparing RNA-seq libraries, using the NEBNext rRNA Depletion Kit v2 and NEBNext Ultra II Directional RNA Library Prep, following the manufacturer's protocol.

## fRIP-seq analysis

The adapter sequences were removed from the reads using Cutadapt (Martin 2011), followed by alignment to the hg19 reference genome using RNA STAR (Dobin et al, 2013). Duplicate reads were removed using UMI-tools deduplicate (Smith et al, 2017). Regions of fRIP-seq IP enrichment were called using MACS2 (Feng et al, 2012; Zhang et al, 2008) with the fRIP-seq input as the control file. The log2FC of fRIP-seq IP/fRIP-seq input was calculated using the aligned BAM file with bamcompare (Ramírez et al, 2016). For visualization purposes, read densities were calculated across the genome as bedgraph files and uploaded onto the IGV genome browser.

## RNA immunoprecipitation assay

The genomic region transcribing TFF1 eRNA was amplified by PCR using the primers listed in Appendix Table S1. The amplified PCR product was then cloned into the pcDNA3.1+ vector downstream of the T7 promoter. The digested vector was subjected to in vitro transcription (IVT) following the manufacturer's protocol using T7 polymerase (Promega P2075) and a biotin RNA labeling mix (Roche 11685597910).

RIP was performed as previously described with minor modifications (Jayani et al, 2017). Briefly, MCF-7 cells were washed and pelleted using 1X PBS. For cell lysis, 1 ml of RIP buffer (25 mM Tris-HCl pH 7.4, 150 mM KCl, 0.5 mM DTT, 0.5% NP-40, 1X protease inhibitor cocktail, and 100 U/ml RNase inhibitor) was added to 5 million cells, followed by 5 min of incubation on ice and ten cycles of sonication using a Bioruptor pico with 30 s ON and 30 s OFF settings. The lysates were then clarified by centrifugation at 12,000 rpm for 12 min. To the cell lysate, 2 µg of IVT-biotin-labeled TFF1 eRNA fragments were added to each tube and incubated on the rotor for 2 h at 4 °C. About 15 µl of Dynabeads MyOne streptavidin beads were added and incubated for an additional 45 min. The beads were washed twice with RIP buffer containing 300 mM KCl, and the proteins were eluted by boiling the beads at 98 °C in 2X laemmli dye.

For the DNA competition RIP assay, unlabeled purified PCR product of the same genomic region was added to the cell lysates in three different conditions: No DNA, 1XDNA concentration, and 2XDNA concentration. For the WT and RBM-ERα RIP assay, ERα WT or RBM constructs were transfected into HEK-293T cells. After 24 h of transfection, the cells were treated with 100 nM E2 and harvested to perform RNA pulldowns.

## 3X ERE biotin pulldowns

To perform ERE pulldowns, 3XERE oligoand its reverse complementary sequence (provided in Appendix Table S2) were annealed, and end labeling was performed using T4 PNK from NEB (cat number: M0201S) and biotin-14-dATP from Invitrogen (cat number: 19524016).

For the ERE pulldowns, cell lysates were either mock-treated or treated with RNase A (Qiagen 19101) and then incubated at 4 °C for 2 h with biotinylated oligos. After 2 h, 20 µl of Dynabeads MyOne streptavidin beads were added and incubated for another 45 min. The beads were then washed three times with NP-40 lysis buffer, and the proteins were eluted by boiling at 98 °C in 2X Laemmli dye.

## For the TFF1 eRNA capturing experiment

HEK-293T cells were transfected using Lipofectamine 2000 according to the manufacturer's protocol with either the empty vector or a vector over-expressing ERα. Cells were then ligand stimulated, and nuclei isolation was performed as previously described (Liu et al, 2014). Briefly, the cell pellet was resuspended in 1 ml of hypotonic buffer (10 mM Tris-HCl pH 7.5, 2 mM MgCl$_2$, 3 mM CaCl$_2$, and 1X protease inhibitor cocktail) and incubated on ice for 5 min, followed by centrifugation at 3000 rpm for 5 min at

4 °C. The swollen cell pellets were lysed by resuspending in lysis buffer (hypotonic buffer + 0.5% NP-40 + 1X protease inhibitor cocktail) and incubated on ice for 5 min. The nuclei were collected by centrifugation at 6000 rpm for 10 min. The nuclei pellet was then resuspended in NP-40 lysis buffer (50 mM Tris-HCl pH 7.4, 150 mM NaCl, 1% NP-40, 0.5% sodium deoxycholate, 0.1% Triton X-100, and 1X protease inhibitor cocktail), incubated on ice for 10 min, and clarified by centrifugation at 12,000 rpm for 12 min. To the 500 µl of nuclear lysates prepared from either the empty vector or ERα overexpressed HEK-293T cells, 0.2 µM of the biotin-labeled 3XERE oligos were added (sequence listed in Appendix Table S2). Unlabeled IVT TFF1 eRNA (Fragment1) was sequentially added to each tube, and the reaction was incubated on a rotor at 4 °C for 2 h. Then, 25 µl of Dynabeads MyOne streptavidin beads were added and incubated for another 45 min. The beads were washed three times with NP-40 lysis buffer, and 1 ml of TRIzol (Invitrogen 15596026) was added to the beads to isolate the RNA. cDNA was prepared, and PCRs were performed using TFF1eRNA.

## Subcellular fractionation

To isolate cytoplasmic, nucleoplasmic, and chromatin-bound proteins, the method described (Caudron-Herger et al, 2019) was adopted. Briefly, cells were swelled and lysed in hypotonic buffer (10 mM Tris-HCl at pH 7.5, 10mMNaCl, 3 mM MgCl₂, 0.3% NP-40, 10% glycerol, 1X protease inhibitor cocktail) by incubating on ice for 10 min. The lysate was then centrifuged at 3000 rpm for 5 min, and the supernatant was transferred to another tube and labeled as the cytoplasmic fraction. The isolated nuclei were washed three times with HLB buffer and then resuspended in modified Wuarin-Schiebler buffer (MWS; 10 mM Tris-HCl at pH 7.5, 300 mM NaCl, 4 mM EDTA, 1 M urea, 1% NP-40, 1% glycerol, 1X protease inhibitor cocktail). For RNAse A treatment of the nucleoplasmic fraction, it was divided into two equal parts, with one part receiving RNAse A at a concentration of 2 µg/ml, while the other part served as the control. The nucleoplasmic fractions were incubated on ice for 10 min. After incubation, the samples were spun at 5000 rpm for 5 min, and the supernatant was labeled as the nucleoplasmic fraction. To isolate the chromatin-bound fraction (CBF), nuclear lysis buffer (NLB: 20 mM Tris-HCl at pH 7.5, 150 mM KCl, 3 mM MgCl₂, 0.3% NP-40, 10% glycerol, 1X protease inhibitor cocktail) was added, followed by sonication for ten cycles with 30 s ON and 30 s OFF settings. To all the fractions, SDS dye was added, and the samples were boiled at 98 °C for 10 min.

## ChIP-seq

MCF-7 cells were seeded and hormone-stripped for 48 h. After that, the cells were transfected with either the WT-ERα or RBM-ERα expressing plasmids. Following 24 h of transfection, the cells were treated with 100 nM E2 for 1 h and crosslinked with 1% formaldehyde for 10 min at room temperature with gentle shaking. The crosslinking reaction was quenched by adding 0.125 M glycine for 5 min at room temperature. ChIP-seq samples were prepared according to the protocol described (Saravanan et al, 2020).

Briefly, the cells were washed thrice with 1X PBS and scraped in 1X PBS. The cell pellets were then spun at 3000 rpm for 5 min to pellet the cells, and the pellets were stored at −80 °C until further use. Approximately 10 million cells were resuspended in 1 ml of

nuclear lysis buffer (NLB) (50 mM Tris-HCl pH 7.4, 1% SDS, 10 mM EDTA pH 8.0, and 1X PIC) and incubated on ice for 10 min. After incubation, the samples were sonicated using a Bioruptor pico for 28 cycles with 30 s ON and 30 s OFF settings. The lysates were then clarified by centrifugation at 12,000 rpm for 12 min. A total of 100 µg of chromatin was measured and diluted 2.5 times with dilution buffer (20 mM Tris-HCl pH 7.4, 100 mM NaCl, 2 mM EDTA pH 8.0, 0.5% Triton X-10, and 1X PIC). A total volume of 500 µl was used for each IP. Additionally, 50 µl was taken out and labeled as 10% input. For each IP sample, 1 µg of either anti-FLAG (Sigma F7425) or anti-PolII (Diagenode C15200004 and sc-55492) antibody was added and incubated overnight at 4 °C on a rotor. For parallel ChIP-seq, both anti-FLAG (Sigma F7425) and anti-CTCF (CST 3418S) antibodies were added simultaneously in the same tube. To pull down the antibody-protein DNA complex, 15 µl of Protein G beads were added the next day and incubated for 4 h at 4 °C on the rotor. After incubation, the beads were washed sequentially with wash buffer I, wash buffer II, wash buffer III, and 1X TE. The DNA-protein complexes were eluted from the beads by resuspending them in 200 µl of elution buffer (100 mM NaHCO3, 1% SDS) at 37 °C in a thermomixer with 1400 rpm. The eluted DNA-protein complexes were reverse-crosslinked by the addition of 14 µl of 5 M NaCl and overnight incubation at 65 °C. The ChIP DNA was purified using phenol:chloroform:isoamyl alcohol and ethanol precipitation. The resulting pellet was resuspended in 10 µl of nuclease-free water (NFW) and used for ChIP-seq library preparation using the NEBNext® Ultra™II DNA Library Prep kit with Sample Purification Beads (E7103L) following the manufacturer's protocol.

## ChIP-seq analysis

The raw reads obtained from the sequencing were aligned to the hg19 reference genome using Bowtie2 (Langmead and Salzberg 2012). Reads with a quality score of less than 20 were filtered out using the Filter Sam or Bam tool (Li et al, 2009). Peaks were called using MACS2 (Feng et al, 2012) with default settings. The resulting peaks were visualized using the IGV genome browser. For pfChIP-seq, the total read counts were plotted for all CTCF peaks and ERα peaks that are not within a 5 kb distance of CTCF peaks.

## RNA knockdown

Short hairpin RNA (shRNA) targeting the 3′UTR of the ESR1 gene, TFF1 enhancer RNA (both sense and antisense strands), and a scramble sequence listed in Appendix Table S4 were cloned into the pLKO.1-TRC cloning vector, which was generously provided by David Root (Addgene plasmid # 10878), following previously described methods (Moffat et al, 2006).

## RNase A ChIP-seq

MCF-7 cells were subjected to E2 stimulation, washed with 1X PBS three times, and pelleted. RNase A ChIP-seq was performed following a previously described protocol (Thakur et al, 2019; Zoabi et al, 2014) with slight modifications. Briefly, the cell pellet was resuspended in a buffer (20 mM HEPES pH 7.5, 0.1 mM CaCl₂, 3 mM MgCl₂, 150 mM NaCl, 0.05% NP-40, and 1X PIC). The cell suspension was divided into two equal parts. One aliquot was treated with RNase A at a concentration of 2 µg/ml, while the other

aliquot was mock-treated. Both aliquots were incubated on ice for 10 min. For the ChIP-seq with pre-extraction, the cells were pelleted by centrifugation at 3000 rpm at 4 °C and washed with the resuspension buffer. The cell pellets were then resuspended in 500 µl of resuspension buffer and crosslinked using 1% formaldehyde for 10 min at room temperature. The crosslinking reaction was quenched with 0.125 mM glycine. For the ChIP-seq without pre-extraction, the cells pellet was resuspended in resuspension buffer (20 mM HEPES pH 7.5, 0.1 mM CaCl$_2$, 3 mM MgCl$_2$, 150 mM NaCl, 0.05% NP-40, and 1X PIC). The suspension was divided into two equal parts, one aliquot was mock-treated, and the other was treated with RNAse A for 10 min on ice. After incubation time 1% formaldehyde was directly added to the cell suspension, followed by quenching with 0.125 mM glycine.

## Generation of ERα-RBM mutant plasmids

The WT-ERα and RNA binding mutants (259–262 RRGG > AAAA) constructs were generated by PCR amplification using the oligos listed in Appendix Table S3. The amplified products of the WT and RBD mutants were then cloned into the 3XFLAG CMV10 vector at the EcoRI and BamHI restriction sites.

## Gradient salt elution for chromatin-bound proteins

Nuclei were isolated from cells expressing either wild-type (WT) or RBM FLAG-tagged ERα following the protocol mentioned above. The isolated nuclei were then resuspended in a buffer containing 10 mM Tris-HCl (pH 7.5), 0.15% NP-40, 2 mM EDTA, 1X PIC, and 75 mM NaCl. The nuclei suspension was rocked at 4 °C for 15 min to allow for protein extraction. After incubation, the nuclei were pelleted by centrifugation at 4000 rpm for 5 min, and the supernatant was collected. This elution step was repeated multiple times, each time using an increasing NaCl concentration. Finally, after the 600 mM NaCl elution, the pellet was resuspended in 2X laemmli dye, and the samples were boiled at 98 °C for 10 min. This process allows for the extraction and collection of proteins associated with the chromatin-bound fraction.

## Retention assay

Cells were grown on coverslips and transfected with ERα:GFP WT or RBM after 48 h of hormone deprivation. After 24 h of transfection, cells were stimulated with E2 for 1 h. The coverslips were washed three times with 1X PBS. For the pre-extraction of weakly associated chromatin-bound proteins, cells were treated with CSK buffer (10 mM PIPES/KOH pH 6.8, 100 mM NaCl, 300 mM sucrose, 1 mM EGTA, 1 mM MgCl$_2$, 1 mM DTT, and 1X PIC) for 5 min on ice. Another set of coverslips were mock-treated to serve as a control. After the incubation, the CSK buffer was removed, and cells were crosslinked using 4% paraformaldehyde. The coverslips were mounted with 90% glycerol. Cell imaging was performed using a 60x/1.42 oil objective on the Olympus FV3000 microscope.

## Dual luciferase assay

MCF-7 cells were grown in a 24-well plate using hormone-deprived media for 48 h. Subsequently, the cells were co-transfected with 250 ng of 3X ERE with TATA-box luciferase plasmid (Addgene 11354), a gift from Donald McDonnell, 2.5 ng of Renilla TK plasmid, and either WT or RBM-ERα. The cells were then treated with E2 for either 24 or 4 h. Luciferase activity was measured using the Multimode Reader Varioskan Lux, with Renilla luciferase used as an internal control. Multiple biological replicates were performed without technical replicates.

## ERE binning

ERα peaks were called using MACS2 (Feng et al, 2012) and motif scores were calculated using FIMO (Grant et al, 2011) with $p$ value less than 1E-4 using the ERE motif from JASPAR database (Matrix ID: MA0112.3). Out of a total of 11,763 peaks, 5311 regions showed the presence of the ERE motif. These regions were divided into five continuous bins based on the decreasing order of the ERE motif score. ERα tag density, Log2F.C. (pBox/WT) and Log2F.C. (RBM/WT), were plotted for these regions.

Similarly, for KLF4 and SOX2, peaks were called using MACS2, and motif scores were calculated using FIMO with the same parameters as before. The motifs were obtained from the JASPAR database, with matrix ID MA0039.2 for KLF4 and MA0143.3 for SOX2. To calculate the motif score, peak coordinates were submitted to FIMO with a $p$ value threshold of less than 0.01. The best motif per site was then selected to determine the motif strength of each category.

## EU-seq

MCF-7 cells were grown in hormone-deprived media for 48 h and transfected with either the WT-ERα or RBM-ERα. After 6 h, the media was changed. Following 24 h of transfection, the cells were treated with 100 nM of E2 for 1 h, during which nascent RNA was labeled with 0.5 mM EU for 45 min. Total RNA was isolated using TRIzol (Invitrogen 15596026) according to the manufacturer's protocol. rRNA depletion was performed using the NEBNext® rRNA Depletion Kit v2 (E7405L). The nascent EU-labeled RNA was biotinylated using biotin azide and captured using Dynabeads MyOne Streptavidin T1 magnetic beads, following the manufacturer's protocol (Click-iT™ Nascent RNA Capture Kit, for gene expression analysis, C10365). Subsequently, cDNA was synthesized using the SuperScript™ IV First-Strand Synthesis System (18091050). The second strand of cDNA was synthesized using the NEBNext® Ultra™ II Directional RNA Library Prep with Sample Purification Beads (Catalog no-E7765L). The double-stranded DNA was further fragmented, and the library was prepared using the NEBNext® Ultra™ II FS DNA Library Prep Kit for Illumina (E7805L), following the manufacturer's protocol.

## EU-seq analysis

The raw reads were aligned to hg19 using Bowtie2 (Langmead and Salzberg 2012). PCR duplicates were removed using MarkDuplicates from Picard Tools (Anon n.d.). Read coverage on known E2 upregulated genes was calculated using multiBamSummary (Ramírez et al, 2016), and the gene list was categorized into changing and unchanging categories.

## FRAP

The FRAP assay was performed after 45 min of E2 (estradiol) treatment using a 488 nm laser. Transfected cells were bleached

with 100% laser power over an area of 1 μm, and images were collected every 2 s after bleaching. Prior to bleaching, five frames were collected. The bleaching process was conducted in the sixth frame, and images were taken for the following 2 min. The intensity of the Region of Interest (ROI) was measured across the time frames. To account for cell-to-cell variability and ROI differences, all the frames were normalized using the mean intensity of the ROI in the first frame. Additionally, the curve was min-max normalized to correct for bleaching differences. For half-time calculation, the time it takes for the normalized intensity to recover to half of its original value was calculated.

## In vitro ERE-based transcription assay

The TFF1 enhancer (chr21:43795658-43798708, assembly hg19) was cloned into the pGL4.23 vector (Luc2 minP) from Promega, upstream to a minimal promoter. The plasmid was linearized by performing restriction digestion using *BamH1* to be used as a template for in vitro transcription. Estradiol-treated MCF-7 cells were used to isolate nuclei as described previously. The isolated nuclei were dissolved in a hypertonic buffer with the following composition: 15 mM HEPES pH 7.6, 400 mM KCl, 5 mM $MgCl_2$, 0.1 mM EDTA, 0.1% Tween 20, 10% Glycerol, and 1 mM DTT (Wibisono and Sun 2021).

For setting up the in vitro transcription assay, 11 μl of the above-prepared nuclear lysate was mixed with 14 μl of transcription buffer (1.5 μl of 50 mM $MgCl_2$, 1.0 μl of 10 mM each rNTPs, 100 μg of template DNA, and 7.5 μl of nuclease-free water). Different concentrations of the mentioned RNA were added to the reaction mixture. The reaction was then incubated at 30 °C for 30 min. The RNA was extracted using TRIzol, followed by cDNA synthesis, and qPCR (Appendix Table S5) was performed to quantify luciferase RNA expression.

## Data availability

The data were deposited in GEO with accession number GSE241216. https://www.ncbi.nlm.nih.gov/geo/query/acc.cgi?acc=GSE241216.

The source data of this paper are collected in the following database record: biostudies:S-SCDT-10_1038-S44318-024-00225-y.

## Peer review information

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

## Acknowledgements

We acknowledge the support of the NCBS-TIFR under the Department of Atomic Energy (Government of India) no. RTI 4006 (to DN). DN is an EMBO Global Investigator (GIN). We also acknowledge the funding support from Welcome-IA (IA/S/23/1/506749) and the DST core grant (CRG/2019/005714) to DN. DS, TK, and RM are supported by the TIFR-NCBS graduate program. We acknowledge the technical support from the core facility at NCBS, specifically NGS and CIFF. DS acknowledges Ananya Sadhu, Awadhesh Pandit, Kuldeep Sachdeva, and Arif Hussain for help in experiments. We thank Sachin Mishra and Amanjot Singh for their critical comments on the manuscript. We thank DN lab members for their discussions.

## Author contributions

**Deepanshu Soota**: Conceptualization; Resources; Data curation; Software; Formal analysis; Validation; Investigation; Visualization; Methodology; Writing—original draft; Project administration; Writing—review and editing. **Bharath Saravanan**: Data curation; Software; Formal analysis; Validation; Visualization; Writing—review and editing. **Rajat Mann**: Data curation; Formal analysis; Methodology; Writing—review and editing. **Tripti Kharbanda**: Methodology; Writing—review and editing. **Dimple Notani**: Conceptualization; Data curation; Supervision; Funding acquisition; Investigation; Writing—original draft; Project administration; Writing—review and editing.

Source data underlying figure panels in this paper may have individual authorship assigned. Where available, figure panel/source data authorship is listed in the following database record: biostudies:S-SCDT-10_1038-S44318-024-00225-y.

## Disclosure and competing interests statement

The authors declare no competing interests.

# Expanded View Figures

**Figure EV1.  ERα interacts with RNA.**

(**A**) Genome browser screenshot showing fRIP-seq IP, Input, and Log2F.C. (IP/Input) for MYC locus. (**B**) Genomic distribution of fRIP-seq peaks differentially enriched either in nucleoplasmic (NP) or chromatin (ca) fraction or distributed equally between chromatin and nucleoplasmic fractions. (**C**) fRIP-PCR using TFF1 eRNA oligos. (**D**) Immunoblot for ERα and GAPDH on RNA pulldowns using biotin-labeled TFF1 eRNA with lysates from cells grown in stripping media treated with either Vehicle or E2. (**E**) Summary plot showing the log2F.C. (fRIP IP/Input) across categories of ERα interacting RNA in Veh and E2 treatment. (**F**) Heatmap depicting ERα intensity on intergenic regions intersecting within 10 kb of caRNA or NPRNA or ca=NPRNA enriched fRIP-seq peaks. (**G**) Plot depicting the distance between the nearest ERα peak and fRIP-seq peak enriched either in caRNA (200 regions), NPRNA (275 regions) or ca=NP (583 regions) RNA. Statistical significance determined by Mann–Whitney $U$-test. Replicates for fRIP-seq and publicly available ERα ChIP-seq are mentioned in Appendix Table S6. (**H**) Illustration explaining the RNA-mediated recruitment of ERα.

▶

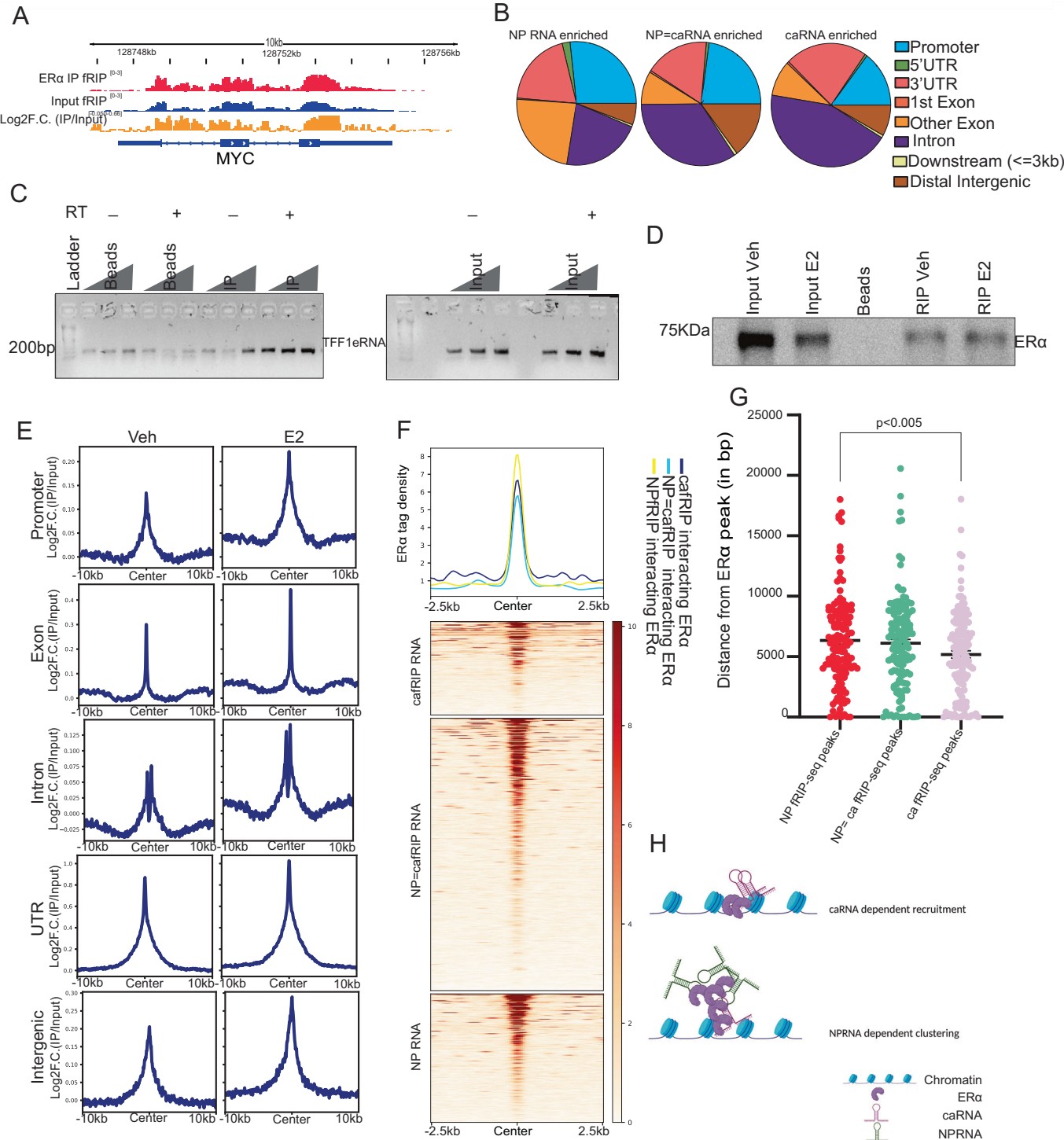

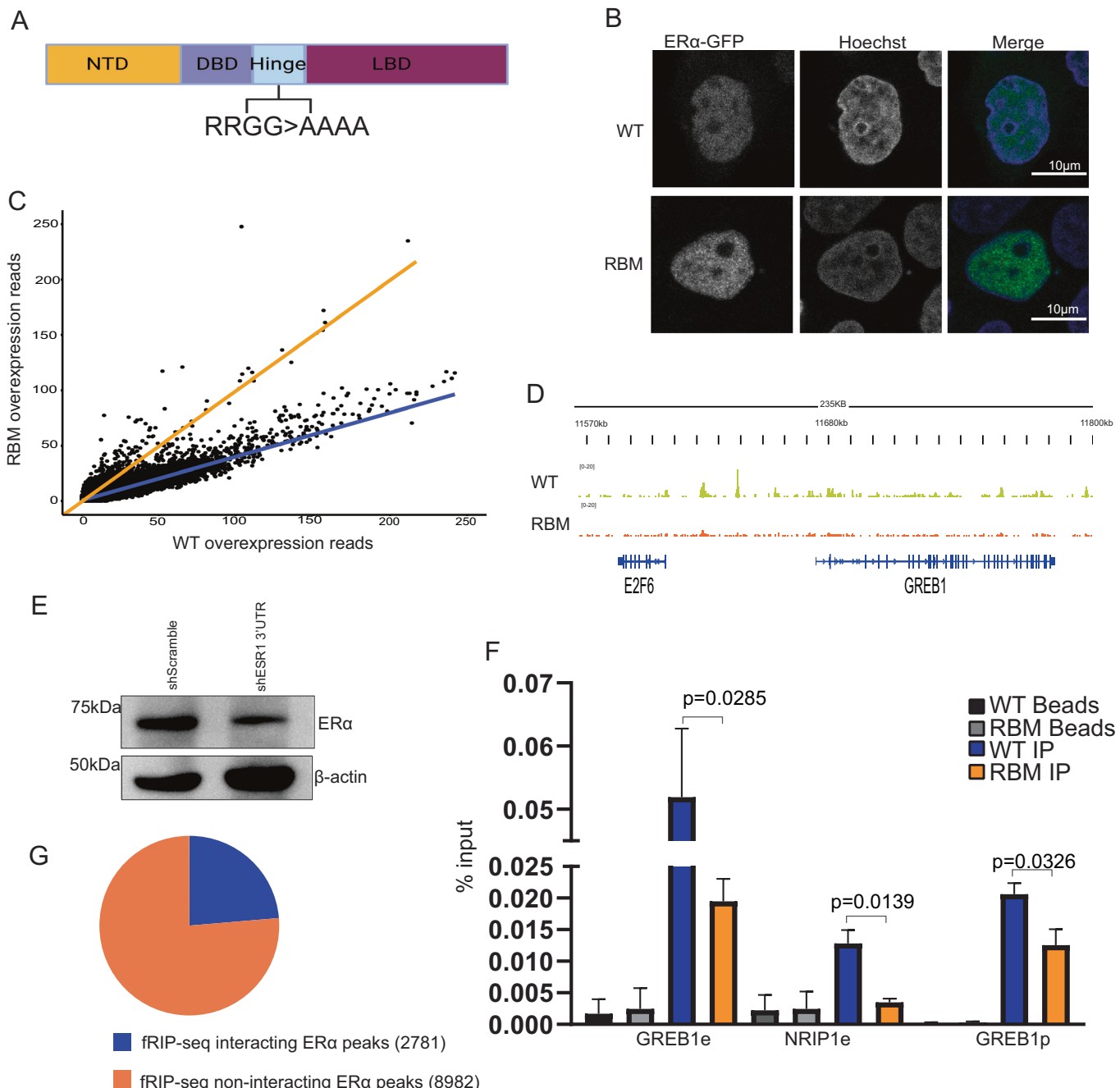

**Figure EV2. RNA binding mutant of ERα shows loss of binding genome-wide.**

(A) Schematic depicting the domains of ERα protein, the RNA binding RRGG sequence, and its mutation to AAAA. (B) Confocal images of ERα:GFP WT and RBM overexpressed in MCF-7 with E2 treatment for 1 h. (C) Normalized read counts showing the distribution of ERα:FLAG WT and RBM ChIP-seq reads. (D) Genome browser snapshot showing ChIP-seq signal of ERα WT and RBM on GREB1 locus in MCF-7. (E) Immunoblot for ERα and β-actin expression in total lysates from MCF-7 cells transfected with short hairpin RNA targeting against either a scramble sequence or the 3'UTR of the ESR1 gene. (F) FLAG enrichment on enhancers of GREB1 and NRIP1, as well as on the promoter of GREB1, in the background of downregulation of endogenous ERα and upon overexpression of WT and RBM FLAG-tagged ERα. Statistical significance was determined by unpaired *t*-test, and error bars denote the standard error of the mean (SEM) with two biological and three technical replicates. (G) Pie chart illustrating the number of ERα peaks categorized based on their interaction with RNA.

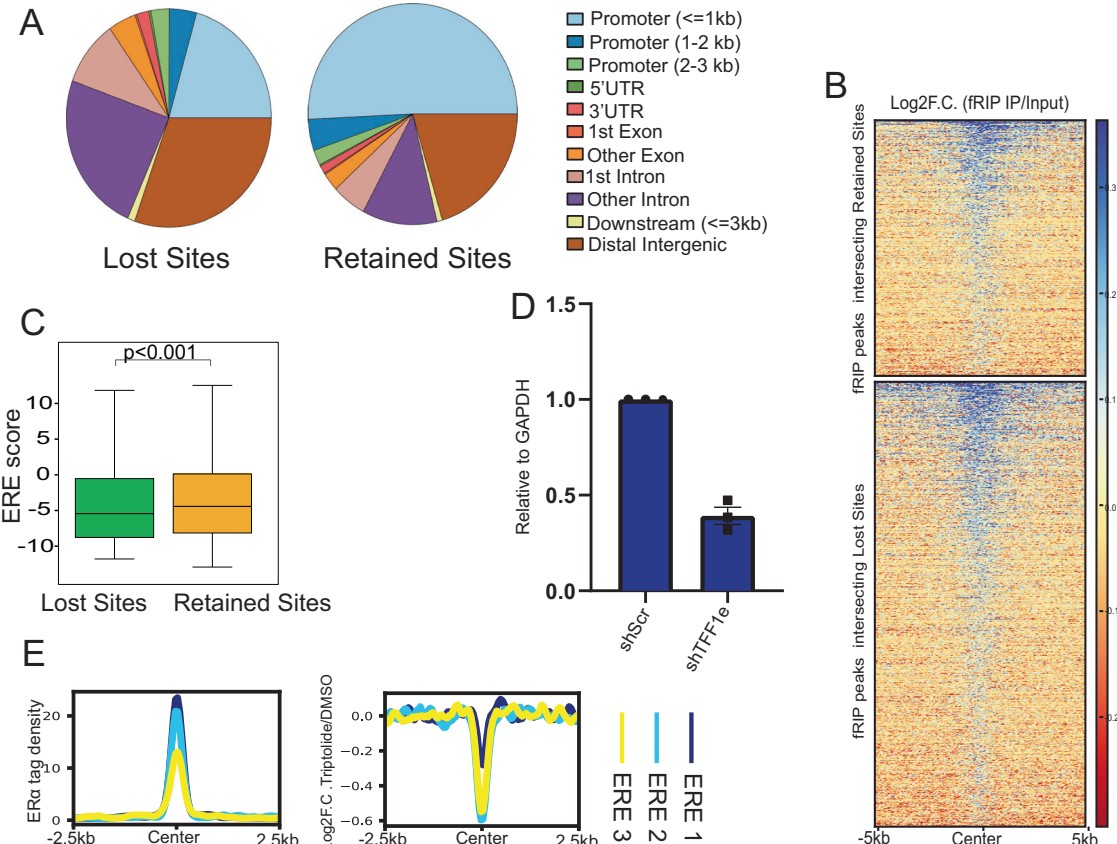

**Figure EV3.  Weaker motifs exhibit higher dependence on RNA for ERα binding.**

(A) Genomic distribution of lost and retained peaks upon overexpression of RBM-ERα over WT-ERα. (B) Log2F.C. IP/Input for the fRIP-seq peaks intersecting with the lost and retained ERα upon RBM-ERα overexpression over WT-ERα. (C) Boxplot depicting the ERE motif score of ERα peaks that are lost (36,587 peaks) and retained (14,668 peaks) upon RBM expression as compared to the WT. Statistical significance determined by Mann–Whitney U-test. (D) qRT-PCR depicts the levels of TFF1 enhancer RNA following shRNA-mediated knockdown compared to scramble. The error bar denotes SEM from three biological replicates. (E) Heatmap depicting ERα tag density and Log2F.C (Triptolide ERα/DMSO ERα) tag density at varying ERE strength. The center lines of the boxplot denote the median, the box limits indicate the 25th and 75th percentiles, and the whiskers extend 1.5 times the interquartile range from the 25th and 75th percentiles. Outliers are not presented.

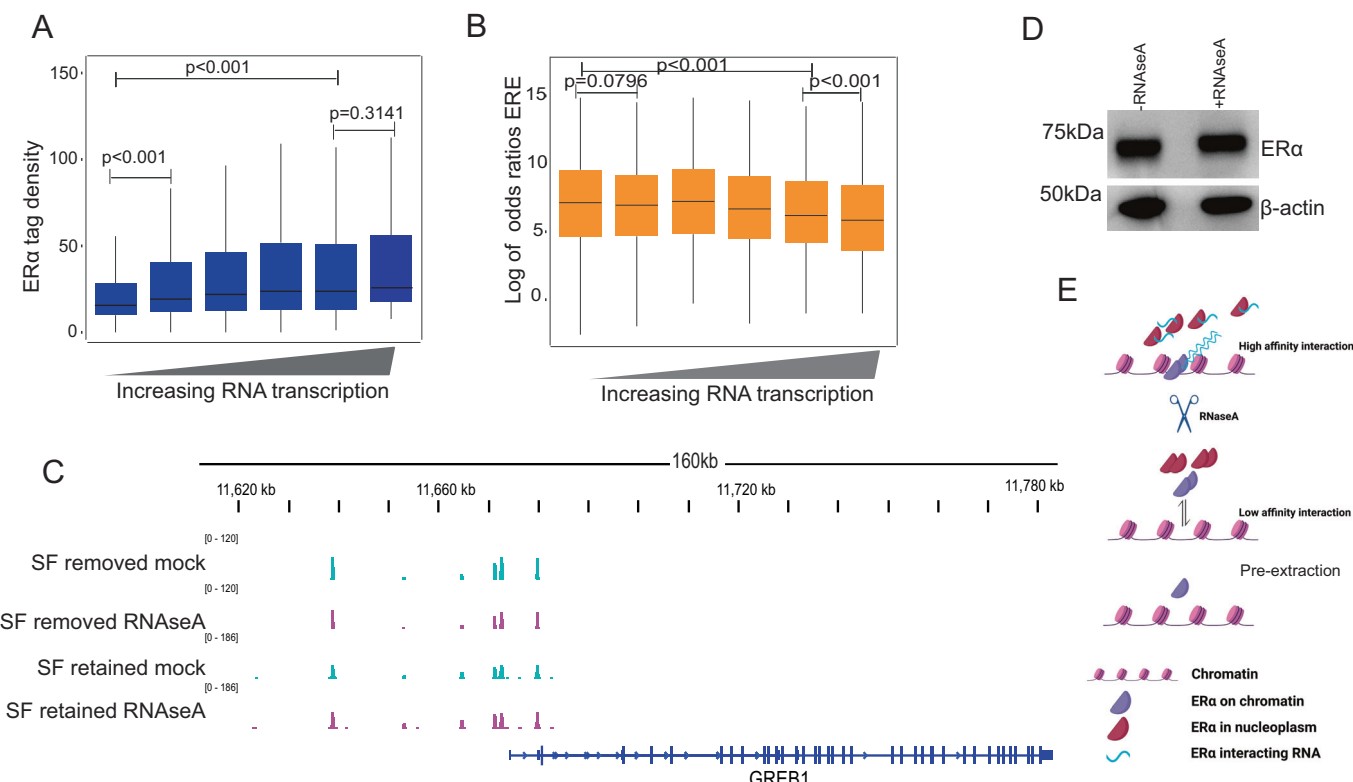

**Figure EV4. ERα retention on chromatin is RNA dependent.**

(A) Boxplot showing ERα enrichment on all sites binned based on the levels of RNA transcription in increasing order. Statistical significance determined by Mann–Whitney *U*-test. (B) Boxplot showing the Log of odds ratio for the ERE motif on all sites binned based on the levels of RNA transcription in increasing order. Statistical significance determined by Mann–Whitney *U*-test. (C) Genome Browser screenshot of GREB1 locus showing ERα ChIP-seq upon RNAse A treatment with removal and retention of soluble proteins. (D) Immunoblot depicting ERα and β-actin levels in MCF-7 isolated nuclei treated with mock or RNAse A. (E) Illustration depicting the RNA is required for ERα on chromatin. The center lines of the boxplot denote the median, the box limits indicate the 25th and 75th percentiles, and the whiskers extend 1.5 times the interquartile range from the 25th and 75th percentiles. Outliers are not presented. In (A, B), the first to sixth bins contains 15,863, 1404, 1988, 1543, 630, and 301 regions, respectively. Replicates for (A, B), publicly available ERα ChIP-seq, and GRO-seq are mentioned in Appendix Table S6.

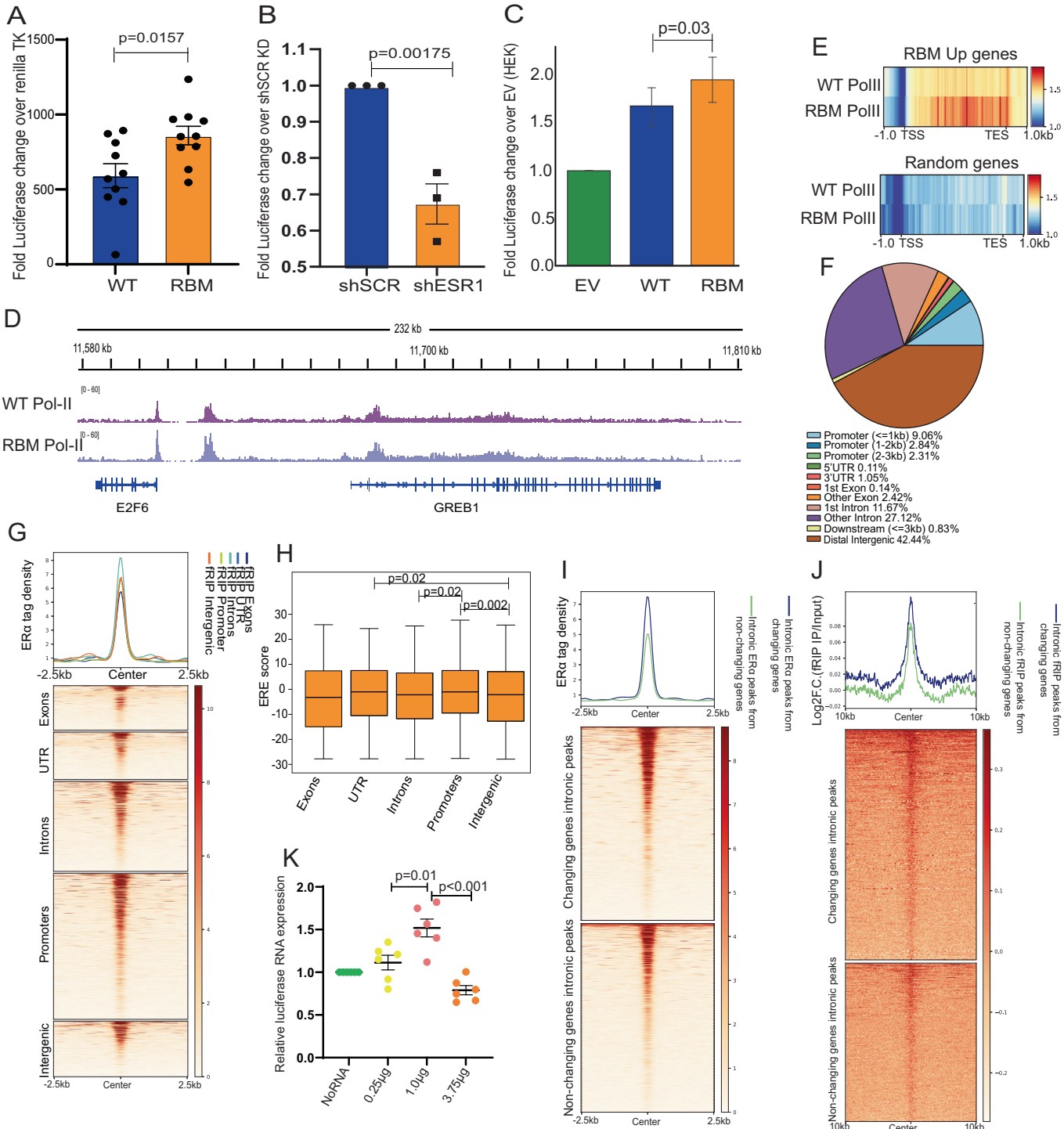

◀ **Figure EV5. The dynamic binding of ERα allows better transcription of target genes.**

(A) 3X ERE-driven firefly luciferase activity normalized to renilla TK luciferase upon 24 h of E2 treatment. Statistical significance determined by Mann–Whitney *U*-test and error bar denotes SEM from five biological replicates with two technical replicates each, plotted all. (B) Luciferase activity normalized to the scramble knockdown for the 3X ERE reporter assay conducted with short hairpin RNA targeting either the scramble sequence or the 3′UTR of the ESR1 gene in MCF-7 cells after 24 h of E2 treatment. Statistical significance was determined using the Mann–Whitney *U*-test, and the error bar denotes SEM from three biological replicates. (C) 3X ERE-driven firefly luciferase activity upon WT-ERα or RBM-ERα overexpression and 24 h of E2 treatment in HEK-293T. Statistical significance determined by Mann–Whitney *U*-test and error bar denotes SEM from three biological replicates. (D) Genome browser screenshot showing the occupancy of total PolII on GREB1 locus upon expression of WT-ERα or RBM-ERα. (E) Profile plot illustrating PolII tag density normalized with respect to *Drosophila* DNA spike in, plotted on genes upregulated by RBM and a random set of genes across the entire gene body. (F) Pie chart depicting the genomic distribution of ERα ChIP-seq peak. (G) Heatmap depicting the ERα intensity across ERα bound regions within 10 kb of various categories of fRIP-seq peaks. (H) Boxplot depicting the ERE motif score from different categories of exonic (445), UTR (486), intronic (943), promoter (1574), and intergenic (574), ERα peaks interacting with RNA. Statistical significance was determined using the Mann–Whitney *U*-test. (I) Heatmap depicting the ERα tag density on intronic sites from changing and non-changing gene categories. (J) Heatmap depicting the fRIP-seq signal from intronic peaks of genes that are changing and non-changing. (K) Relative luciferase RNA expression from an in vitro transcription reaction with varying concentrations of added RNA. Statistical significance is determined by unpaired *t*-test and error bar denotes SEM from three biological replicates. The center lines of the boxplot denote the median, the box limits indicate the 25th and 75th percentiles, and the whiskers extend 1.5 times the interquartile range from the 25th and 75th percentiles. Outliers are not presented.

