## [Peer Review File · The EMBO Journal]

RNA fine-tunes estrogen receptor-alpha binding on low-affinity DNA motifs for transcriptional regulation

Deepanshu Soota, Bharath Saravanan, Rajat Mann, Tripti Kharbanda, and Dimple Notani

Corresponding author(s): Dimple Notani (dnotani@ncbs.res.in)

Review Timeline:

Submission Date:	29th Aug 23
Editorial Decision:	28th Nov 23
Revision Received:	2nd Apr 24
Editorial Decision:	31st May 24
Revision Received:	12th Jun 24
Accepted:	15th Jul 24

Editor: *Cornelius Schneider*

Transaction Report:

Dear Dr. Notani,

Thank you for submitting your manuscript for consideration by the EMBO Journal and for sharing a preliminary revision plan with me.

Based on the positive evaluation of the manuscript by referees #1 and #2 and your willingness to engage in a major revision as indicated during the pre-decision consultation, I would like to invite you to submit a revised version of the manuscript, addressing the comments of all three reviewers as indicated in the preliminary point-by-point reply. I should add that it is EMBO Journal policy to allow only a single round of revision, and acceptance of your manuscript will therefore depend on the completeness of your responses in this revised version. If you have any additional questions or want to discuss the revisions further, I am happy to do so by email or video conferencing.

Thank you for the opportunity to consider your work for publication. I look forward to your revision.

Yours sincerely,

Cornelius Schneider, PhD
Editor
The EMBO Journal
c.schneider@embojournal.org

Please remember: Digital image enhancement is acceptable practice, as long as it accurately represents the original data and conforms to community standards. If a figure has been subjected to significant electronic manipulation, this must be noted in the

figure legend or in the 'Materials and Methods' section. The editors reserve the right to request original versions of figures and the original images that were used to assemble the figure.

We realize that it is difficult to revise to a specific deadline. In the interest of protecting the conceptual advance provided by the work, we recommend a revision within 3 months (26th Feb 2024). Please discuss the revision progress ahead of this time with the editor if you require more time to complete the revisions. Use the link below to submit your revision:

Referee #1:

The manuscript by Soota et al. studies the influence of RNA binding on the dynamics of chromatin binding and the transcriptional activity of ligand-activated Estrogen Receptor-alpha (ER α). The manuscript presents a novel evidence that underscores the role of RNA as a negative feedback mechanism in modulating the transcriptional potential of ER α . Briefly, authors identify the myriad RNA molecules with which ER α interacts during E2 signaling. The involvement of RNA on strength of ER α binding on weaker motif was shown by the combination of biochemical fractions using RNaseA and RNA-binding deficient mutant of ER α . By testing the DNA-binding deficient mutant, authors show the dependence of stronger motifs on DBD of ER α whereas, the weaker motifs depend on RNA binding.

Furthermore, using FRAP, nascent RNA-seq and Pol2 ChIP-seq, they observe that RNA interactions fine tune ligand dependent activation response of ER α to achieve an optimal level of transcription. Absence of RNA binding heightens gene activation through regulation of ER α dynamics on chromatin leading to increased loading of RNA Polymerase II. In Summary, this study pioneers in demonstrating how RNA assists TF in stable binding on chromatin leading to robust transcription in initial phase of ligand stimulation, but attenuates the transcriptional potential at the later time of ligand stimulation.

The Oksuz et al have also studied the implication of RNA:TF interaction but have not demonstrated the effect of RNA binding on ER α mediated transcription upon ligand stimulation. I congratulate the Soota et al for thorough and rigorous experimental setup leading to these novel observations.

I have following specific comments that would require experiments:

- 1) In Figures 1 and 1S the authors illustrate the RNAs that interact with ER α under estrogen stimulation including those from distal intergenic regions and introns etc. Additionally, they demonstrate that some ER α -bound sites interact with RNA while others do not. Authors should explore the distinguishing features of these ER α -interacting binding sites.
- 2) In Figures 3 and 3S, the authors demonstrate a genome-wide correlation between RNA and ER α occupancy, revealing that ER α -binding sites with a similar motif strength tend to recruit more ER α due to the presence of RNA. Furthermore, they integrate biochemical fractionation with ChIP-seq and illustrate that the removal of total RNA results in a weaker ER α interaction with chromatin. In the experiment involving pre-extraction of loosely bound proteins, ER α loss is observed. Is the degree of loss in the genome, dependent on RNA in the same categories as in Figure 3A? Additionally, when authors do not perform pre-extraction, they observe a gain of binding. Under such circumstances, is the gain of binding solely dependent on the ERE motif?
- 3) In Figure 4D, the authors demonstrated that the RNA-binding mutant (RBM) of ER α interacts weakly with chromatin. To support this data, authors should perform similar experiments with WT-type ER α in presence of RNaseA. Does ER α elute from chromatin at a lower salt concentration under these conditions? This data is crucial in ruling out the non-specific effects of mutant.
- 4) Moreover, in the representative images of WT and RBM-ER α GFP (Fig. S2B and Fig. 4A), differences in nuclear distribution are apparent. Since, ER α is known to phase separate, do authors find any differences in condensates between WT and RBM mutant?
- 5) Some sections in the manuscript need more clarity in writing. Overall, the manuscript would improve from english editing.

Referee #2:

Soota et al performed an extensive analysis of how RNA binding can regulate ER's dynamic vs static activity in breast cancer cells. RNA binding increases ER binding but stability and the loss of RNA binding leads to ER binding loss, but it activates ER dynamic activity, hence transcription. This is interesting, but also confusing at various results. The results show that the RNA binding is essential for low strength EREs, but these show high transcription and H3K27ac activity. The study also shows that the promoters were less susceptible to the loss of RNA binding. This means that eRNAs and other non-promoter based RNAs

from low strength EREs are playing a major role in stabilising ER binding. However, only few results are focused on this aspect and generally the effect are shown global (whenever chromatin fractionation or so is involved). I think this needs to be further differentiated (also in the abstract), otherwise the summary of the results is quite confusing and contradictory. The low affinity/low strength EREs show the dependency with ER surrounding non-promoter based RNAs is the key finding, and this must be highlighted better.

More concerns:

- According to this study, eRNA expression should lead to more stable ER binding and less transcription. High eRNA binding is highly correlated with higher transcription of gene promoters. However, this is not really the case. Deeper investigation of this is necessary.
- Fig. 1F-G showing more ER-TFF1 eRNA interactions after E2 treatment, is this not because more expression of eRNAs, more eRNAs- more pulldown?
- Low strength EREs are very few. For example, the transition in the regulation of transcription or H3K27ac occupancy is better seen from ER2-5 and it's a lost in ER6. May be the clusters can be joined to make this clear. Few EREs in a cluster brings the average disrupted comparing to other clusters. Random number of sites can be explored to avoid the bias. The intensity is also looking similar between clusters due to the three-color scale (blue-white-orange) heatmap format. This could be much easier to interpret with two color-scale heatmap eg., white to green or white to orange or so.
- Fig. S3A clusters show low strength EREs show higher transcription. Fig. 3A shows high transcription is related with high ER binding. This is quite contradictory. May be focusing on promoter and enhancer-based signals separately might make it clear.
- RNase A treatment reduces ER binding to the chromatin. How is the ER expression after RNase A treatment.
- Relate all the investigated genes/EREs/ luciferase enhancer if they are from low strength EREs. TFF1, GREB1 - are these in the low strength EREs?
- Fig. 5B - "set of upregulated genes" from EU-seq data in cells expressing RBM-ER compared to WT-ER. What are these genes? Do they belong to low strength EREs regulated genes?
- Using RNase A will kill all the RNAs. How specific is this regulation with eRNAs? What happens if you inhibit only few eRNA expression specifically?

Minor concerns:

- Results of Fig. 1H needs to be explained more clearly. What is the point of using unlabelled DNA, this won't be pulled down anyway.
- Check the x-axis of Figure 2F. "50" is missing and "0" is moved in.
- Overall, the three scale heatmaps are so confusing to understand when they are compared. Fig. 2G - very difficult to get the heatmaps together to understand this due to the difference in colour scales of two different heatmaps. Log2FC - brighter orange means better regulation, GRO-seq- brighter orange means less signal. Its better to keep this consistent to understand that they are indirectly proportional.

Referee #3:

In the study "RNA binding negatively regulates ligand induced transcriptional potential of estrogen receptor-alpha (ER α)" by Soota et al. the authors study the interaction of ER α with various RNAs via its RNA binding motif in the hinge region. The authors suggest that the RNA binding mutant RBM exhibits reduced ER α binding genome wide as well as higher nuclear mobility in vivo as demonstrated by photobleaching experiments. The authors also claim that in comparison to the WT ER α , acute estrogen stimulation induced more "robust" polymerase loading and transcription of ER α -regulated genes in cells overexpressing the RBM mutant. The authors conclude that ER α interactions with RNA increase the spatial confinement of the receptor, which negatively impacts its ability to activate estrogen-induced genes. Most of the presented data agree with the previously published findings by Xu et al. 2021 and extend them by providing data on the in vivo dynamics of the RBM; however, the two studies come to opposing conclusions regarding the transcriptional effects of the RBM mutant on ER α -regulated genes.

Overall, while the RBM mutant is potentially interesting, the authors conclusions are not supported by the presented data. This could be, in part, due to the presence of endogenous WT ER α which will interfere with any observable phenotype from the over-expression of the mutant protein. I would encourage the authors to downregulate endogenous ER α and then overexpress their WT and RBM for future studies.

I outline some of my major concerns below and recommend that this manuscript be rejected at this time.

1. Comparison with Xu et al, 2021:

The authors' explanation that Xu et al. studied the role of the RBM mutant under "basal" condition that does not provide ER activation is however unfounded. A quick look at the culture conditions in the Xu et al. paper shows that the cells were grown in media supplemented with 10% FBS that contains more than enough estrogen to activate the ER. For example, in the Rodriguez et al. 2019 study (cited here as well), media with complete FBS was used to activate the ER and upregulate the TFF1 gene.

2. Weak or modest effects on RNA-binding and transcription:

- The transcription data presented in the manuscript (except the luciferase experiment in Fig. 5A) show modest effects of the RBM mutant and for such differences to be considered significant, additional controls (for example spike-in controls) will be necessary.

- For the RIP-seq experiments, the IP/input fold change is very weak (-0.05 to -0.66 log₂ fold change) and does not show significant enrichment over background.
- In Fig 1G, the GAPDH level is substantially higher for E2 as compared to input Vehicle which weakens one of the central arguments of the manuscript: "ligand strengthens ER α :RNA interactions (Fig. 1G and Fig. S1D)" (lines 122-123)
- The authors should calculate the fold change in enrichment {plus minus} E2 rather than only in the treatment condition. Heatmaps should also show both vehicle and E2 treatment in all figures.

3. Switching between cell lines and experimental conditions:

- The authors switch back and forth between HEK293 and MCF-7 cells with no explanation of why they do this and without explicitly stating which cell line was used for which experiment (in the Results section).
- Different experiments were done with different amount of hormone deprivation (72 h for RIP-seq vs 48 h for ChIP-seq).

As presented, this study does not provide enough evidence that "RNA interactions spatially confine ER α which negatively impacts the ligand-dependent transcriptional upregulation of estrogen-induced genes."

Referee #1 (Report for Author)

The manuscript by Soota et al. studies the influence of RNA binding on the dynamics of chromatin binding and the transcriptional activity of ligand-activated Estrogen Receptor-alpha ($ER\alpha$). The manuscript presents a novel evidence that underscores the role of RNA as a negative feedback mechanism in modulating the transcriptional potential of $ER\alpha$. Briefly, authors identify the myriad RNA molecules with which $ER\alpha$ interacts during E2 signaling. The involvement of RNA on strength of ER α binding on weaker motif was shown by the combination of biochemical fractions using RNaseA and RNA-binding deficient mutant of ER α . By testing the DNA-binding deficient mutant, authors show the dependence of stronger motifs on DBD of ER α whereas, the weaker motifs depend on RNA binding.

Furthermore, using FRAP, nascent RNA-seq and Pol2 ChIP-seq, they observe that RNA interactions fine tune ligand dependent activation response of ER α to achieve an optimal level of transcription. Absence of RNA binding heightens gene activation through regulation of ER α dynamics on chromatin leading to increased loading of RNA Polymerase II. In Summary, this study pioneers in demonstrating how RNA assists TF in stable binding on chromatin leading to robust transcription in initial phase of ligand stimulation, but attenuates the transcriptional potential at the later time of ligand stimulation.

The Oksuz et al have also studied the implication of RNA:TF interaction but have not demonstrated the effect of RNA binding on ER α mediated transcription upon ligand stimulation. I congratulate the Soota et al for thorough and rigorous experimental setup leading to these novel observations. I have following specific comments that would require experiments:

We thank the reviewer for finding our work interesting and recognizing the novelty of our data and rigorous nature of work.

1) In Figures 1 and 1S the authors illustrate the RNAs that interact with $ER\alpha$ under estrogen stimulation including those from distal intergenic regions and introns etc. Additionally, they demonstrate that some $ER\alpha$ -bound sites interact with RNA while others do not. Authors should explore the distinguishing features of these $ER\alpha$ -interacting binding sites

1. As pointed out by the reviewer, our data indicate that weaker ERE motifs are dependent on RNA for stable $ER\alpha$ binding (Fig. 3A-B, S3C, and S3E). Additionally, using publicly available data, we observed that other RNA binding TF such as KLF4 and SOX2 also bind on weaker motifs due to their RNA binding ability, we have provided this new data in Fig. 3C-D with description in lines 179-181.

2. Furthermore, concerning the genomic identity of these regions, our findings suggest that RNA predominantly interacts with $ER\alpha$ at intronic and intergenic regions in a ligand-dependent manner (Fig. 1A, S1B, S1E), resulting in the stabilization of $ER\alpha$ at these sites (Fig. 4, S4E, and S5G). Notably, these regions are well-documented as active enhancers (Li et al., 2013; Saravanan et al., 2020). Consequently, we show that these intergenic and intronic regions lose more $ER\alpha$ upon RBM expression compared to promoters (Fig. 2I-J and S3A). Together, data suggest that enhancers exhibit a greater dependence on RNA compared to promoters (Fig. 1B and S3B). This data is described in lines 154-157.

3. Functionally, highly induced genes upon signaling (changing genes (Fig. 6H and 6J)) exhibited higher binding of $ER\alpha$ in RNA dependent manner as compared to non-changing genes (Fig. S5I-J). Upon RBM expression, these sites exhibit a greater loss of $ER\alpha$ compared to non-changing genes (Fig.6K), indicating that RNA on these highly active sites stabilizes $ER\alpha$ (Fig. 6L), thereby negatively impacting gene expression (Fig. 6G and 6I).

We thank reviewer for emphasizing on the role of RNA in recruiting ER α on weaker EREs and, that RNA assists in gene transcription through such non-canonical mechanisms. We have accordingly changed the abstract and title to reflect this new insight.

2) In Figures 3 and 3S, the authors demonstrate a genome-wide correlation between RNA and ER α occupancy, revealing that ERE-binding sites with a similar motif strength tend to recruit more ER α due to the presence of RNA. Furthermore, they integrate biochemical fractionation with ChIP-seq and illustrate that the removal of total RNA results in a weaker ER α interaction with chromatin. In the experiment involving pre-extraction of loosely bound proteins, ER α loss is observed. Is the degree of loss in the genome, dependent on RNA in the same categories as in Figure 3A? Additionally, when authors do not perform pre-extraction, they observe a gain of binding. Under such circumstances, is the gain of binding solely dependent on the ERE motif?

We have now performed the analysis of ER α ChIP-seq with and without RNase A to test if in the absence of RNA, the binding of ER α can be predicted based on the strength of ERE's. The analysis clearly suggests that when the total RNA is depleted, ER α binding becomes weaker in RNA dependent manner (Fig. 4L). The data is mentioned in lines 208-210. However, if allowed to rebind in absence of RNA, the ER α binding is driven by strength of the ERE motif (Fig. 4M). The lines 212-214 describe this data.

3) In Figure 4D, the authors demonstrated that the RNA-binding mutant (RBM) of ER α interacts weakly with chromatin. To support this data, authors should perform similar experiments with WT-type ER α in presence of RNaseA. Does ER α elute from chromatin at a lower salt concentration under these conditions? This data is crucial in ruling out the non-specific effects of mutant.

In accordance with the reviewer's suggestion, we have conducted salt elution of proteins from cellular lysate in the presence or absence of RNaseA (Fig. 5G). Our observations reveal that ER α binding with chromatin is low-affinity upon RNaseA treatment. This observation is similar to the other observation that RBM-ER α has low affinity binding on chromatin as compared to WT-ER α (Fig. 5F). The description of this new data is added in the lines 236-239 of revised manuscript

4) Moreover, in the representative images of WT and RBM-ER α GFP (Fig. S2B and Fig. 4A), differences in nuclear distribution are apparent. Since, ER α is known to phase separate, do authors find any differences in condensates between WT and RBM mutant?

We thank the reviewer for this insightful suggestion, we have conducted ER α quantitative imaging using both the WT and RBM-ER α :GFP and observed that the normalized area fraction occupied by RBM-ER α GFP is less as compared to the WT-ER α (Fig. 5D). The data suggests that RNA mediated stable binding of ER α on chromatin assists in robust phase separation whereas, the dynamic binding of mutant ER α negatively affects phase separation. The data is in agreement to the previous reports showing RNA assisted phase separation and condensate maturation for TDP43 and FXR1 (Grese et al., 2021; Smith et al., 2020). The description of this new data is added in the lines 228-231 of revised manuscript.

However, in the RNA-binding domain mutant, this RNA-mediated crosslinking is compromised, resulting in dynamic condensates and consequently, transcriptional response increases. These results are consistent with recent reports indicating that stable phase-separated condensates can inhibit the transcriptional response (Chen et al., 2023; Chong et al., 2022).

5) Some sections in the manuscript need more clarity in writing. Overall, the manuscript would improve from english editing.

We have made efforts to enhance the overall clarity and quality of the language in the revised manuscript.

Referee #2 (Report for Author)

Soota et al performed an extensive analysis of how RNA binding can regulate ER's dynamic vs static activity in breast cancer cells. RNA binding increases ER binding but stability and the loss of RNA binding leads to ER binding loss, but it activates ER dynamic activity, hence transcription. This is interesting, but also confusing at various results. The results show that the RNA binding is essential for low strength EREs, but these show high transcription and H3K27ac activity. The study also shows that the promoters were less susceptible to the loss of RNA binding. This means that eRNAs and other non-promoter based RNAs from low strength EREs are playing a major role in stabilising ER binding. However, only few results are focused on this aspect and generally the effect are shown global (whenever chromatin fractionation or so is involved). I think this needs to be further differentiated (also in the abstract), otherwise the summary of the results is quite confusing and contradictory. The low affinity/ low strength EREs show the dependency with ER surrounding non-promoter based RNAs is the key finding, and this must be highlighted better.

We thank the reviewer for insightful comments on our work and suggestion that we should look deeper in to the relationship of ERE strength and their dependence on RNA for ER α binding. Further, correlate this data with transcriptional output. We have now performed new experiments and data analysis to understand this relationship better. As a result, we have included a new Figure (Fig. 3) that entirely focuses on ERE strength and RNA dependence (details below). We thank the reviewer for suggesting these experiments and data comparisons which in our view, has strengthen the manuscript.

Weaker ERE's depend on RNA for ER α binding:

As pointed out by the reviewer, our data indicate that weaker ERE motifs are dependent on RNA for stable ER α binding (Fig. 3A-B, S3C, and S3E). In order to test the generality of this claim, in the revised manuscript, we have added data on other RNA interacting TF, KLF4 and SOX2. The CHIP-seq data comparing WT and RBM mutants to these two proteins suggest that weaker motifs are most affected if RNA binding domain of KLF4 and SOX2 is mutated (Fig. 3C-D). This new data is described at lines 179-181.

Furthermore, concerning the genomic identity of these regions that rely on RNA for ER α binding, our findings suggest that RNA predominantly interacts with ER α at intronic and intergenic regions in a ligand-dependent manner (Fig. 1A, S1B, S1E). Therefore, RBM-ER α shows maximum loss at these sites whereas, the promoters were least affected (Fig. 2I-J and S3A). As per reviewer's suggestion, we observed weaker EREs in intergenic and intronic regions as compared to the promoters (Fig. S5H) (description in lines 268-270). Together, data suggests that intergenic and intronic regions exhibit more binding with RNA, they harbour weaker EREs and require RNA for ER α binding at these sites.

Weaker ERE's are transcriptionally more active:

We have previously shown that intergenic and intronic regions harbour active enhancers bound by ER α (Li et al., 2013; Saravanan et al., 2020). Further, E2-regulated genes are driven by intergenic and intronic enhancers that are known as Megatrans. These megatrans enhancers are experimentally validated to be superior and show high mediator and polymerase loading as compared to other ER α -bound enhancers (Liu et al., 2014). In order to test if enhancer function, motifs and RNA has any relationship, we compared megatrans enhancers with other high ER α -bound regions (rebuttal Fig. 1A). We observed that megatrans enhancers have less ER α as opposed to other highly occupied ER α -bound regions (rebuttal Fig. 1B). Further, megatrans enhancers harbor weaker EREs (rebuttal Fig. 1C). Despite weaker ERE, the

transcriptional potential of megatrans enhancers was significantly higher than the other category as revealed by high eRNA transcription from these enhancers (rebuttal Fig. 1D).

It's noteworthy that previous work by Bas Ven Steensel's group has also demonstrated that high-affinity p53 DNA sites are less transcriptionally potent than middle-affinity ones (Trauernicht et al., 2023)

We appreciate reviewers suggestions to explore the relationship of weaker EREs with transcription. The data, that most active enhancers harbour weaker but not stronger EREs is very exciting. Further, the active enhancers do not stably bind with ER α suggests that low affinity binding sites with low TF binding are perhaps selected evolutionary for induced transcription. However, these claims would require substantial further investigation, including CRISPR-mediated changes in motif strength to assess how alterations in motif strength affect target gene transcription and ER α binding. We will pursue this as an independent study. For this reason, we have not included the rebuttal Fig. 1 in the main manuscript. We hope that the reviewer will understand and agree with our reasoning.

As the reviewer also pointed that low affinity/ low strength EREs at non-promoter regions show the dependency on RNA is the key finding of our work, and this must be highlighted better, we have included Figure 3.

We thank reviewer for emphasizing on the role of RNA in recruiting ER α on weaker EREs in intergenic/intronic vs. promoter regions. We have accordingly changed the abstract and title to reflect this new insight.

Rebuttal Figure 1: Comparison of Megatrans enhancers with high ER α binding sites: (A) The overlap between high ER α -bound regions and megatrans. (B) Heatmap depicting the ER α binding strength on megatrans and high-ER α bound regions. (C-D) ERE score and eRNA transcription levels from GRO-seq comparing two categories of ER α -bound regions.

More concerns:

- According to this study, eRNA expression should lead to more stable ER binding and less transcription. High eRNA binding is highly correlated with higher transcription of gene promoters. However, this is not really the case. Deeper investigation of this is necessary.

We agree with the reviewer that high eRNA levels are well-correlated with increased transcription from the cognate promoter, a relationship we previously reported in the context of estrogen-induced signaling (Li et al., 2013, Saravanan et al., 2020). Therefore, the claim that ER α :RNA interaction negatively regulates transcription may seem confusing.

Our data suggest that low levels of RNA/eRNA that is present on genomic sites under basal signaling, help in ER α binding which would lead to transcriptional activation by recruiting polymerase and other activators. However, as the act of transcription produces more and more eRNA/RNA. RNA:ER α crosslinking makes ER α and perhaps other proteins like p300 and mediators, PolIII around these enhancers, less dynamic. As a result, this stable binding down regulates the transcription. This is why RBM:ER α which does not allow the crosslinking at least between RNA and ER α , this negative modulation of the transcriptional response cannot be achieved. Therefore, genes regulated by the weaker EREs in intronic and intergenic regions are further upregulated.

To support this claim, we have performed two new experiments:

1. An in vitro reporter assay using increasing concentration of RNA. Which suggests that while low levels of RNA support transcription, high levels inhibit it (Fig. S5K). The data is described in lines 281-286 and in discussion section at lines 357-361.

2. ER α forms phase-separated condensates in response to E2 stimulation (Boija et al., 2018; Nair et al., 2019; Saravanan et al., 2020). In order to understand if the high transcriptional potential seen in case of ER α -RBM is also recapitulated by the dynamic condensate behaviour which is linked with higher transcriptional rate, we quantified ER α -condensate volume comparing WT and RBM-ER α . This new data indeed shows that mutant occupies less condensate volume due to dynamic mobility of RBM-ER α as revealed by FRAP (Fig. 5D). The data is described in lines 228-231. Similar observations that stable phase-separation may inhibit the transcription has been shown (Chen et al., 2023; Chong et al., 2022; Guan et al., 2019; Kim et al., 2022).

- Fig. 1F-G showing more ER -TFF1 eRNA interactions after E2 treatment, is this not because more expression of eRNAs, more eRNAs- more pulldown?

The reviewer's comment was prompted by a lack of experimental detail in the manuscript, we regret the inconvenience. To provide clarity, eRNA was in vitro transcribed using biotin-rNTP, and an equal concentration was utilized as bait on streptavidin beads. To test if liganded ER α has different affinity for RNA, we loaded MCF-7 lysates made from cells that were either treated or untreated with estrogen. In conclusion, equal RNA was taken as bait. We have now included more details on experimental procedure in the method section at lines 456-472.

- Low strength EREs are very few. For example, the transition in the regulation of transcription or H3K27ac occupancy is better seen from ER2-5 and it's a lost in ER6. May be the clusters can be joined to make this clear. Few EREs in a cluster brings the average disrupted comparing to other clusters. Random number of sites can be explored to avoid the bias. The intensity is also looking similar between clusters due to the three-color scale (blue-white-orange) heatmap format. This could be much easier to interpret with two color-scale heatmap eg., white to green or white to orange or so.

We thank reviewer for this suggestion. We have combined ER α bins 5 and 6, resulting in equal size of all 5 bins (Fig. 3A). Three-color heatmap are now plotted in two colors (Fig 3A-B).

- Fig. S3A clusters show low strength EREs show higher transcription. Fig. 3A shows high transcription is related with high ER binding. This is quite contradictory. May be focusing on promoter and enhancer-based signals separately might make it clear.

We regret for this confusion. The analyses presented in Figure 3A and S3A were conducted on different regions. Figure S3A (now Fig. 3A-B and rebuttal Fig. 2) demonstrate that lower strength EREs have higher H3K27ac, PolII and transcription. In Figure 3A (now Fig. 4A-B), the purpose was to bin sites with similar strength ERE in to bins of increasing transcription from them to test if RNA has any role in ER α binding. Indeed, the results showed increasing ER α occupancy with level of RNA indicative of RNA dependent stabilization of ER α despite similar motif strength.

Notably, the highest transcribing bin (6th), similar to the lower strength ERE motif analysis, displayed a lower motif score (Fig. 4B), less ER α tag density (Fig. 4A) but the highest transcription. This is consistent with the observations that lower strength EREs exhibit higher transcription. We have elaborated on these results for better clarity at lines 187-193.

Rebuttal Figure 2: EREs binning into higher to lower score strength bins show that lower strength bins exhibit more H3K27ac (A), PolII (B) and nascent transcription (C).

- RNase A treatment reduces ER binding to the chromatin. How is the ER expression after RNase A treatment?

We have performed immunoblotting of ER α in samples treated with and without RNase A in the revised manuscript. The levels of ER α remain constant with and without RNase A treatment. This data has been added as Fig. S4D with description in lines 214-215.

- Relate all the investigated genes/EREs/ luciferase enhancer if they are from low strength EREs. TFF1, GREB1 - are these in the low strength EREs?

We thank the reviewer for this insightful comment.

Our data on signalling and other recent studies (in vitro and in vivo) from colleagues in the field point that weaker and dynamic binding of TF correlates with better transcriptional output (Buchanan et al., 2001; Chen et al., 2023; Chong et al., 2022; Guan et al., 2019; Kim et al., 2022). In this regard, we observed that most active enhancers exhibit weaker motifs as compared to other enhancers with high ER α binding (rebuttal fig. 1). Further, active enhancer element with weaker motifs exhibit high H3K27ac (rebuttal Fig. 2A), PolII (rebuttal Fig. 2B), and nascent RNA transcription (rebuttal Fig. 2C). Therefore, we posit that the capacity of an ER α binding site to function as an enhancer is greater with a weaker motif, owing to its dynamic interactions with DNA.

Further, as shown previously that megatrans enhancers exhibit weaker ERE motifs as opposed to high ER-bound sites regions (rebuttal Fig. 1C). Despite weaker ERE, the transcriptional potential of megatrans enhancers was significantly higher than the other category (rebuttal Fig. 1D). These findings suggest that weaker ERE motifs on enhancers have greater transcriptional potential. Notably, the genes upregulated upon RBM- ER α (changing genes) such as TFF1, GREB1 and CCND1 are the target of megatrans with weaker EREs. We have added this text in result section at lines 271-273

Fig. 5B - "set of upregulated genes" from EU-seq data in cells expressing RBM-ER compared to WT-ER. What are these genes? Do they belong to low strength EREs regulated genes?

The genes in changing category were all closer to megatrans enhancers (intergenic or intronic). As shown in rebuttal Fig. 1, megatrans enhancers have weaker motifs as compared to other high ER α -binding enhancers. The genes upregulated upon RBM- ER α (changing genes) such as TFF1, GREB1 and CCND1 are the target of megatrans with weaker EREs. We have added this text in result section at lines 271-273.

- Using RNase A will kill all the RNAs. How specific is this regulation with eRNAs? What happens if you inhibit only few eRNA expression specifically?

We performed knockdown of sense and anti-sense eRNA from the enhancer of TFF1 gene by shRNAs (Fig. S3F). The ChIP of ER α on TFF1 enhancer but not FOXC1 enhancer (Negative control) shows specific reduction of ER α (Fig. 3E). This finding aligns with a previous report, which demonstrated that tethering of specific eRNA to endogenous loci results in higher ER α occupancy (Hou and Kraus, 2022). We have included this new data as Fig. 3E and have described it in lines 181-183.

Minor concerns:

- Results of Fig. 1H needs to be explained more clearly. What is the point of using unlabelled DNA, this won't be pulled down anyway.

We regret the inconvenience caused due to lack of clarity. Since, DNA and RNA both bind with ER α , we used cold DNA as a competitor to ER α :RNA complex to test if DNA can compete for ER α that is already bound to RNA. We have added details in the revised manuscript at lines 111-114.

- Check the x-axis of Figure 2F. "50" is missing and "0" is moved in.

We thank reviewer for pointing out this mislabelling.

- Overall, the three scale heatmaps are so confusing to understand when they are compared. Fig. 2G - very difficult to get the heatmaps together to understand this due to the difference in colour scales of two different heatmaps. Log2FC - brighter orange means better regulation, GRO-seq- brighter orange means less signal. Its better to keep this consistent to understand that they are indirectly proportional.

We agree with the reviewer. We have replaced the heatmaps in the manuscript.

Referee #3 (Report for Author)

In the study "RNA binding negatively regulates ligand induced transcriptional potential of estrogen receptor-alpha (ER α)" by Soota et al. the authors study the interaction of ER α with various RNAs via its RNA binding motif in the hinge region. The authors suggest that the RNA binding mutant RBM exhibits reduced ER α binding genome wide as well as higher nuclear mobility in vivo as demonstrated by

photobleaching experiments. The authors also claim that in comparison to the WT ER α , acute estrogen stimulation induced more "robust" polymerase loading and transcription of ER α -regulated genes in cells overexpressing the RBM mutant. The authors conclude that ER α interactions with RNA increase the spatial confinement of the receptor, which negatively impacts its ability to activate estrogen-induced genes. Most of the presented data agree with the previously published findings by Xu et al. 2021 and extend them by providing data on the in vivo dynamics of the RBM; however, the two studies come to opposing conclusions regarding the transcriptional effects of the RBM mutant on ER α -regulated genes.

1. Overall, while the RBM mutant is potentially interesting, the authors conclusions are not supported by the presented data. This could be, in part, due to the presence of endogenous WT ER α which will interfere with any observable phenotype from the over-expression of the mutant protein. I would encourage the authors to downregulate endogenous ER α and then overexpress their WT and RBM for future studies.

We thank reviewer for suggesting this important control experiment. We downregulated endogenous ER α using shRNA designed against the 3'UTR of the ESR1 mRNA. This approach resulted in significant downregulation of ER α at the protein levels (Fig. S2E), description in lines 146-148. We then tested the effectiveness of ER α removal on ERE-driven reporter assay. The reporter activity was reduced upon endogenous ER α knockdown (Fig. S5B), confirming the downregulation of endogenous ER α . To compare the transcriptional effects of WT and RBM-ER α , we then expressed WT or RBM- ER α in the background of reduced endogenous ER α and performed reporter assays. The RBM-ER α showed higher transcriptional ability as compared to WT-ER α (Fig 6B). The data is described in lines 248-252.

Next, under the depletion of endogenous ER α , we extended the finding from ChIP-seq that RBM-ER α shows less binding with chromatin (Fig. 2). We compared the occupancy of WT and RBM-ER α at GREB1 enhancer and its promoter, as well as on NRIP1 enhancer. Similar to ChIP-seq, we observed reduction of RBM-ER α binding as compared to WT-ER α (Fig. S2F). This new data is described in lines 146-150.

Taken together, our results suggest that RBM-ER α binding on chromatin is less but shows greater transcriptional potential in response to ligand.

I outline some of my major concerns below and recommend that this manuscript be rejected at this time.

1. Comparison with Xu et al, 2021:

The authors' explanation that Xu et al. studied the role of the RBM mutant under "basal" condition that does not provide ER activation is however unfounded. A quick look at the culture conditions in the Xu et al. paper shows that the cells were grown in media supplemented with 10% FBS that contains more than enough estrogen to activate the ER. For example, in the Rodriguez et al. 2019 study (cited here as well), media with complete FBS was used to activate the ER and upregulate the TFF1 gene.

We disagree with the reviewer, normal growth media has a fairly low concentration of estrogen, approximately 2 pg/mL of estradiol in 10% FBS, compared to physiological levels of 15–350 pg/mL of estradiol in the plasma of premenopausal women (Jang et al., 2023). Meaning, 10% complete media in cell culture only provides basal if not less than basal levels of estrogen. For these reasons, as a standard practice in the field, 10 to 100nM E2 is provided to cells that have been stripped for serum for 72h for signaling stimulation (Brown et al., 1984, Li et al., 2013, Saravanan et al., 2020, Liu et al., 2014).

Therefore, complete media experimental setting in Xu et al reflect the signalling under concentration comparable to 2pg/mL, not E2 stimulation conditions. Further, Rodriguez et al., used complete media and compared the results with E2 treatments. For estrogen stimulation, they also used charcoal stripped serum media like our study.

Further, we are providing evidence that E2-regulated genes respond to estrogen signaling as compared to basal signaling in rebuttal Fig. 3. We compared the differential expression of a set of genes that are upregulated in case of RBM-ER α expression with random genes using GRO-seq in MCF-7 cells grown in complete media (GSM3675886) vs. estrogen-treated conditions (GSM4259520). The RBM-ER α upregulated genes described in our study exhibited higher fold changes compared to a randomly selected set of genes upon E2 stimulation (rebuttal Fig. 3). This finding suggests that the estrogen-regulated program is dependent on estrogen stimulation, and these genes respond to estrogen stimulation as compared to complete media.

To summarise, Xu et al and our study are done under distinct estrogenic environment therefore, the gene transcription effects of E2-induced genes are different.

Rebuttal Figure 3: Estrogen stimulation over basal signaling shows the upregulation of genes that were sensitive to ER α -RBM expression.

2. Weak or modest effects on RNA-binding and transcription:

- The transcription data presented in the manuscript (except the luciferase experiment in Fig. 5A) show modest effects of the RBM mutant and for such differences to be considered significant, additional controls (for example spike-in controls) will be necessary.

As suggested by the reviewer, we employed a drosophila spike-in controlled Pol2 ChIP-seq approach following the expression of WT and RBM-ER α for efficient normalization of the data. The spike-in normalised signal at RBM-ER α upregulated and random genes is consistent with our previous findings that PolII is more enrichment on genes in case of RBM-ER α (Fig. S5E). The lines 258-260 describe this new data in the revised manuscript.

ChIP-seq suggesting RBM-ER α has weaker binding in the genome was already done with CTCF-ChIP as spike in. We used Co-ChIP with ER α and CTCF because CTCF does not change upon ligand stimulation (Holding et al., 2018). Notably, we found that the RBM-ER α signal specifically decreased in genome while CTCF levels remained unchanged (Fig. 2E-F)

We have shown the increased transcriptional potential of mutant ER α by reporter assays, PolII ChIP-seq, EU-seq and chromatin fractionation. All these assays corroborate with each other that mutant has better transcriptional potential. We have found these effects to be statistically significant as mentioned on the graphs.

Further, within EU-seq, we have compared different categories of genes (E2 regulated and non-regulated) and within these E2 regulated genes, the only genes where ER α binds in the gene body show the statistically significant upregulation upon mutant expression. The fact that not all E2- regulated gene are changing, suggests that internal backgrounds are similar in WT and RBM-ER α experimental setup.

- For the RIP-seq experiments, the IP/input fold change is very weak (-0.05 to -0.66 log₂ fold change) and does not show significant enrichment over background.

In the genome browser screenshot and heatmaps, the log₂FC scale ranges from -0.05 to 0.66 and log₂FC 0.30, respectively. These scales align with previous fRIP-seq studies for other transcription factors like SOX2, where log₂FC typically falls within the range of 0-1 (Holmes et al., 2020). To further substantiate our fRIP-seq results, we conducted fRIP-qPCR, and in vitro RNA pulldown experiments, presenting the evidence in Figure S1.

We are not making novelty claim about ER α :RNA interaction in our study as this has been shown by multiple studies including Xu et al. (2021) and Oksuz et al. (2023). Furthermore, Steiner et al. (2022) demonstrated in vitro binding of RNA with the hinge region of ER α . These collective results including ours provide robust validation for the observed ER α :RNA interaction with and without ligand stimulation.

- In Fig 1G, the GAPDH level is substantially higher for E2 as compared to input Vehicle which weakens one of the central arguments of the manuscript: "ligand strengthens ER α :RNA interactions (Fig. 1G and Fig. S1D)" (lines 122-123)

The GAPDH levels in the immunoblot are different because of intentional loading of unequal amount of lysates taken, to achieve equal levels of ER α in unliganded and liganded samples. This is because, upon E2 treatment, the levels of ER α decrease (Kocanova et al., 2010; Saravanan et al., 2020; Totta et al., 2014), a well-known ligand-receptor degradation phenomenon. We also observe similar E2 dependent reduction in ER α levels as shown here (rebuttal Fig. 4).

To accurately assess how ER α :TFF1 eRNA interaction changes between unliganded and liganded ER α , it was crucial to maintain similar ER α levels. Using equal GAPDH for normalization would lead to less ER α in the liganded sample.

In response to this concern, we have provided the immunoblot for TFF1 eRNA RIP with an equal amount of lysate from vehicle and E2 treatment (Fig. S1D), description in lines 109-110. The results indicate that there is less ER α protein in the E2-treated lysate. Despite this, the amount of ER α being pulled down with TFF1 eRNA is similar or slightly higher, suggesting that, although the lysate contains less ER α compared to the vehicle, the pulldown is more.

Moreover, ligand induced interactions of ER α :RNA are also supported by fRIP-seq in minus and plus E2 conditions (Supp Fig 1). The comparison between them is already presented in Fig 1SE.

Rebuttal Figure 4: Immunoblotting of ER α (upper panel) and GAPDH (lower panel) on estrogen-treated and untreated cellular lysates.

- The authors should calculate the fold change in enrichment {plus minus} E2 rather than only in the treatment condition. Heatmaps should also show both vehicle and E2 treatment in all figures.

The comparison between minus and plus E2 conditions is depicted in Figure S1D.

Calculating the Log₂FC of fRIP-seq IP/input is essential as the RNA levels of E2-target genes increases upon signaling. Log₂FC with input normalization ensures that we are reporting the actual read counts between E2 and vehicle conditions, thereby eliminating any potential bias arising from differences in input read counts.

3. Switching between cell lines and experimental conditions:

- The authors switch back and forth between HEK293 and MCF-7 cells with no explanation of why they do this and without explicitly stating which cell line was used for which experiment (in the Results section).

We regret for not providing details in results section though, the cell line use was mentioned in the material and methods section. We have now added the reasons of using 293T for reporter assay in results section in lines 128-130 and 252-253.

All experiments were performed in MCF-7 cells. In order to rule out the effect of endogenous ER α , we performed reporter assays and RNA immunoprecipitations in HEK293 as well as MCF-7 because HEK293 does not express ER α . To conclude, we have replicated the same data in HEK293 to test the robustness of the result.

- Different experiments were done with different amount of hormone deprivation (72 h for RIP-seq vs 48 h for ChIP-seq).

We have followed the standard 72 hours stripping protocol for all the experiments. We have now made changes in text at lines 395-415 for better clarity.

For the ChIP-seq, we hormone-stripped the cells for 48 hours, then transfect and allow another 24 hours of stripping to achieve a total of 72 hours of stripping. After this period, a 1-hour E2 stimulation was provided. Similar, 72h of total stripping was done in fRIP-seq. We reinstate that all experiments were done with 72h of stripping.

As presented, this study does not provide enough evidence that "RNA interactions spatially confine ER α which negatively impacts the ligand-dependent transcriptional upregulation of estrogen-induced genes."

RNA spatially confines TF has been shown using single particle tracking by Robert Tijan lab for CTCF (Hansen et al., 2019) and Richard Young group for KLF4, SOX2, GATA2 and RUNX (Oksuz et al., 2023). Our study builds on this prior knowledge to reveal that it is the weaker motifs that require RNA for TF binding and this interaction then leads to regulated ligand-dependent transcription where the dynamic movement of ER α is favourable.

ER α forms phase-separated condensates in response to E2 stimulation (Nair et al., 2019, Boija et al., 2019, Saravanan et al., 2020). In order to understand if the high transcriptional potential seen in case of ER α -RBM is also recapitulated by the dynamic condensate behaviour which is linked with higher transcriptional rate, we quantified ER α -condensate volume comparing WT and RBM-ER α . This new data indeed shows that mutant occupies less condensate volume due to dynamic mobility of RBM-ER α as revealed by FRAP (Fig. 5D). The data is described in lines 228-231. Similar observations that stable phase-separation may inhibit the transcription has been shown (Chen et al., 2023; Chong et al., 2022; Guan et al., 2019; Kim et al., 2022).

Together, these observations support the single particle tracking results from Bob Tijan and Richard Young groups. We hope that addition of imaging data will be convincing to the reviewer.

References:

- Boija, A., Klein, I.A., Sabari, B.R., Dall'Agnesse, A., Coffey, E.L., Zamudio, A.V., Li, C.H., Shrinivas, K., Manteiga, J.C., Hannett, N.M., Abraham, B.J., Afeyan, L.K., Guo, Y.E., Rimel, J.K., Fant, C.B., Schuijers, J., Lee, T.I., Taatjes, D.J., Young, R.A., 2018. Transcription Factors Activate Genes through the Phase-Separation Capacity of Their Activation Domains. *Cell* 175, 1842-1855.e16. <https://doi.org/10.1016/j.cell.2018.10.042>
- Brown, A.M.C., Jeltsch, J.-M., Roberts, M., Chambon, P., 1984. Activation of pS2 Gene Transcription is a Primary Response to Estrogen in the Human Breast Cancer Cell Line MCF-7. *Proc. Natl. Acad. Sci. U. S. A.* 81, 6344–6348.
- Buchanan, G., Yang, M., Harris, J.M., Nahm, H.S., Han, G., Moore, N., Bentel, J.M., Matusik, R.J., Horsfall, D.J., Marshall, V.R., Greenberg, N.M., Tilley, W.D., 2001. Mutations at the Boundary of the Hinge and Ligand Binding Domain of the Androgen Receptor Confer Increased Transactivation Function. *Mol. Endocrinol.* 15, 46–56. <https://doi.org/10.1210/mend.15.1.0581>
- Chen, L., Zhang, Z., Han, Q., Maity, B.K., Rodrigues, L., Zboril, E., Adhikari, R., Ko, S.-H., Li, X., Yoshida, S.R., Xue, P., Smith, E., Xu, K., Wang, Q., Huang, T.H.-M., Chong, S., Liu, Z., 2023. Hormone-induced enhancer assembly requires an optimal level of hormone receptor multivalent interactions. *Mol. Cell* 83, 3438-3456.e12. <https://doi.org/10.1016/j.molcel.2023.08.027>
- Chong, S., Graham, T.G.W., Dugast-Darzacq, C., Dailey, G.M., Darzacq, X., Tjian, R., 2022. Tuning levels of low-complexity domain interactions to modulate endogenous oncogenic transcription. *Mol. Cell* 82, 2084-2097.e5. <https://doi.org/10.1016/j.molcel.2022.04.007>
- Grese, Z.R., Bastos, A.C., Mamede, L.D., French, R.L., Miller, T.M., Ayala, Y.M., 2021. Specific RNA interactions promote TDP-43 multivalent phase separation and maintain liquid properties. *EMBO Rep.* 22, e53632. <https://doi.org/10.15252/embr.202153632>
- Guan, J., Zhou, W., Hafner, M., Blake, R.A., Chalouni, C., Chen, I.P., De Bruyn, T., Giltneane, J.M., Hartman, S.J., Heidersbach, A., Houtman, R., Ingalla, E., Kategaya, L., Kleinheinz, T., Li, J., Martin, S.E., Modrusan, Z., Nannini, M., Oeh, J., Ubhayakar, S., Wang, X., Wertz, I.E., Young, A., Yu, M., Sampath, D., Hager, J.H., Friedman, L.S., Daemen, A., Metcalfe, C., 2019. Therapeutic Ligands Antagonize Estrogen Receptor Function by Impairing Its Mobility. *Cell* 178, 949-963.e18. <https://doi.org/10.1016/j.cell.2019.06.026>
- Hansen, A.S., Hsieh, T.-H.S., Cattoglio, C., Pustova, I., Saldaña-Meyer, R., Reinberg, D., Darzacq, X., Tjian, R., 2019. Distinct Classes of Chromatin Loops Revealed by Deletion of an RNA-Binding Region in CTCF. *Mol. Cell* 76, 395-411.e13. <https://doi.org/10.1016/j.molcel.2019.07.039>

- Hou, T.Y., Kraus, W.L., 2022. Analysis of estrogen-regulated enhancer RNAs identifies a functional motif required for enhancer assembly and gene expression. *Cell Rep.* 39. <https://doi.org/10.1016/j.celrep.2022.110944>
- Hsieh, C.-L., Fei, T., Chen, Y., Li, T., Gao, Y., Wang, X., Sun, T., Sweeney, C.J., Lee, G.-S.M., Chen, S., Balk, S.P., Liu, X.S., Brown, M., Kantoff, P.W., 2014. Enhancer RNAs participate in androgen receptor-driven looping that selectively enhances gene activation. *Proc. Natl. Acad. Sci. U. S. A.* 111, 7319–7324. <https://doi.org/10.1073/pnas.1324151111>
- Jang, S.-H., Paek, S.H., Kim, J.-K., Seong, J.K., Lim, W., 2023. A New Culture Model for Enhancing Estrogen Responsiveness in HR+ Breast Cancer Cells through Medium Replacement: Presumed Involvement of Autocrine Factors in Estrogen Resistance. *Int. J. Mol. Sci.* 24, 9474. <https://doi.org/10.3390/ijms24119474>
- Kim, S., Au, C.C., Jamalruddin, M.A.B., Abou-Ghali, N.E., Mukhtar, E., Portella, L., Berger, A., Worroll, D., Vatsa, P., Rickman, D.S., Nanus, D.M., Giannakakou, P., 2022. AR-V7 exhibits non-canonical mechanisms of nuclear import and chromatin engagement in castrate-resistant prostate cancer. *eLife* 11, e73396. <https://doi.org/10.7554/eLife.73396>
- Kocanova, S., Mazaheri, M., Caze-Subra, S., Bystricky, K., 2010. Ligands specify estrogen receptor alpha nuclear localization and degradation. *BMC Cell Biol.* 11, 98. <https://doi.org/10.1186/1471-2121-11-98>
- Li, W., Notani, D., Ma, Q., Tanasa, B., Nunez, E., Chen, A.Y., Merkurjev, D., Zhang, J., Ohgi, K., Song, X., Oh, S., Kim, H.-S., Glass, C.K., Rosenfeld, M.G., 2013. Functional roles of enhancer RNAs for oestrogen-dependent transcriptional activation. *Nature* 498, 516–520. <https://doi.org/10.1038/nature12210>
- Liu, Z., Merkurjev, D., Yang, F., Li, W., Oh, S., Friedman, M.J., Song, X., Zhang, F., Ma, Q., Ohgi, K.A., Krones, A., Rosenfeld, M.G., 2014. Enhancer Activation Requires trans-Recruitment of a Mega Transcription Factor Complex. *Cell* 159, 358–373. <https://doi.org/10.1016/j.cell.2014.08.027>
- Nair, S.J., Yang, L., Meluzzi, D., Oh, S., Yang, F., Friedman, M.J., Wang, S., Suter, T., Alshareedah, I., Gamliel, A., Ma, Q., Zhang, J., Hu, Y., Tan, Y., Ohgi, K.A., Jayani, R.S., Banerjee, P.R., Aggarwal, A.K., Rosenfeld, M.G., 2019. Phase separation of ligand-activated enhancers licenses cooperative chromosomal enhancer assembly. *Nat. Struct. Mol. Biol.* 26, 193–203. <https://doi.org/10.1038/s41594-019-0190-5>
- Oksuz, O., Henninger, J.E., Warneford-Thomson, R., Zheng, M.M., Erb, H., Vancura, A., Overholt, K.J., Hawken, S.W., Banani, S.F., Lauman, R., Reich, L.N., Robertson, A.L., Hannett, N.M., Lee, T.I., Zon, L.I., Bonasio, R., Young, R.A., 2023. Transcription factors interact with RNA to regulate genes. *Mol. Cell.* <https://doi.org/10.1016/j.molcel.2023.06.012>
- Saravanan, B., Soota, D., Islam, Z., Majumdar, S., Mann, R., Meel, S., Farooq, U., Walavalkar, K., Gayen, S., Singh, A.K., Hannenhalli, S., Notani, D., 2020. Ligand dependent gene regulation by transient ER α clustered enhancers. *PLOS Genet.* 16, e1008516. <https://doi.org/10.1371/journal.pgen.1008516>
- Smith, J.A., Curry, E.G., Blue, R.E., Roden, C., Dundon, S.E.R., Rodríguez-Vargas, A., Jordan, D.C., Chen, X., Lyons, S.M., Crutchley, J., Anderson, P., Horb, M.E., Gladfelter, A.S., Giudice, J., 2020. FXR1 splicing is important for muscle development and biomolecular condensates in muscle cells. *J. Cell Biol.* 219, e201911129. <https://doi.org/10.1083/jcb.201911129>
- Totta, P., Pesiri, V., Marino, M., Acconcia, F., 2014. Lysosomal function is involved in 17 β -estradiol-induced estrogen receptor α degradation and cell proliferation. *PloS One* 9, e94880. <https://doi.org/10.1371/journal.pone.0094880>
- Trauernicht, M., Rastogi, C., Manzo, S.G., Bussemaker, H.J., van Steensel, B., 2023. Optimisation of TP53 reporters by systematic dissection of synthetic TP53 response elements. *Nucleic Acids Res.* 51, 9690–9702. <https://doi.org/10.1093/nar/gkad718>
- Tsai, P.-F., Dell’Orso, S., Rodriguez, J., Vivanco, K.O., Ko, K.-D., Jiang, K., Juan, A.H., Sarshad, A.A., Vian, L., Tran, M., Wangsa, D., Wang, A.H., Perovanovic, J., Anastasakis, D., Ralston, E., Ried, T., Sun, H.-W., Hafner, M., Larson, D.R., Sartorelli, V., 2018. A Muscle-Specific Enhancer RNA

Mediates Cohesin Recruitment and Regulates Transcription in *Trans. Mol. Cell* 71, 129-141.e8.
<https://doi.org/10.1016/j.molcel.2018.06.008>

Dear Dr Notani,

Thank you for submitting a revised version of your manuscript. Referees #1 and #2 find that their concerns were addressed and now recommend publication of the manuscript. Unfortunately, referee #3 did not respond to our request to re-review the manuscript and we have consulted with a fourth referee. This referee thinks that you have adequately responded to all the concerns raised by referee #3 and recommends publication of your manuscript.

There remain only a few mainly editorial points that have to be addressed before I can extend formal acceptance of the manuscript:

1. FUNDING INFO: missing in eJP: NCBS-TIFR; TIFR-NCBS graduate program; core facility at NCBS specifically NGS and CIFF
2. Please add up to 5 keywords
3. COI: missing, should be named "DISCLOSURE AND COMPETING INTERESTS STATEMENT"
4. AC/CRedit: section needs to be removed
5. FIGURE CALLOUTS: there is a callout for Fig. 3F (no such panel) but missing for Fig. 3E; missing callouts for: Fig. 4C-D; TableS5 and TableS6
6. Figures in separate files: yes, but Supplementary figures should be renamed to Figure EV1-EV5, figure labels, legends and callouts
7. APPENDIX 1 FILE WITH ToC: Appendix file needs to be in PDF format; nomenclature should be Appendix Figure S1, S2... and Appendix Table S1, S2...; title page with ToC with page numbers is missing; "Supp. Figures and Tables" and "List of primers and list of tables" should be combined into one Appendix PDF
8. Synopsis:
Papers published in The EMBO Journal are accompanied online by a 'Synopsis' to enhance discoverability of the manuscript. It consists of A) a short (1-2 sentences) summary of the findings and their significance, B) 3-4 bullet points highlighting key results and C) a synopsis image that is 550x300-600 pixels large (width x height, jpeg or png format). You can either show a model or key data in the synopsis image. Please note that the image size is rather small and that text needs to be readable at the final size. Please send us this information together with the revised manuscript.
9. There are 2 sets of appendix figures upld.
(Supp. Figures and Tables) Please choose one version.
10. Please note that the box plots need to be defined in terms of minima, maxima, centre, bounds of box and whiskers, and percentile in the legends of figures 2h, j; 3b, d; 4a-b, l-m; 5c-e; 6g-k, supplementary figures 3c; 4a-b; 5h.
11. Please note that information related to n is missing in the legends of figures 2h, j; 3b, d; 4a-b, l-m; 5c-e; 6a-b, g-k, supplementary figures 1g; 3c-d; 4a-b; 5a-c, h, k.
12. Please note that n=2 in supplementary figure 2f.
13. Although 'n' is provided, please describe the nature of entity for 'n' in the legends of figure 3e, supplementary figure 2f.
14. Please note that the error bars are not defined in the legends of figures 6a-b, supplementary figures 3d; 5a-c, k."
15. Please note that the specific URL for GSE241216 dataset is not provided in the data availability statement.
16. Figure Legends (main + EV): Please note that the legend for figure 4m is incorrectly labelled as 4k.
17. Please indicate the statistical test used for data analysis in the legends of figures 2e-f; 4f-g, supplementary figure 5k.
18. Please note that the red empty circles are not defined in the legend of figure 5a. This needs to be rectified.

With best regards,

Cornelius Schneider

Cornelius Schneider, PhD
Editor | The EMBO Journal
c.schneider@embojournal.org

- a point-by-point response to the referees' comments, with a detailed description of the changes made (as a word file).

- a word file of the manuscript text.

- individual production quality figure files (one file per figure)

- a complete author checklist, which you can download from our author guidelines

(<https://www.embopress.org/page/journal/14602075/authorguide>).

- Expanded View files (replacing Supplementary Information)

We realize that it is difficult to revise to a specific deadline. In the interest of protecting the conceptual advance provided by the work, we recommend a revision within 3 months (29th Aug 2024). Please discuss the revision progress ahead of this time with the editor if you require more time to complete the revisions. Use the link below to submit your revision:

Referee #1:

The authors have addressed all the concerns that I have raised during my first review and this manuscript has been greatly approved. This manuscript has provided extensive data to demonstrated the effect of RNA binding on ERa mediated transcription upon ligand stimulation. I believe it will of great significance to the gene regulation field. I support to publish this manuscript.

Referee #2:

Thanks for considering all the comments very positive and addressing them by providing details meticulously. I am very happy

with the revised version, except the figure 2G, where the heatmap is changed according to my comments, but the colour scale key is not changed matching the new version heatmap.

Referee #4:

I was asked to specifically comment on the responses to Reviewer 3's concerns, since this reviewer has not responded.

In my opinion, the authors have comprehensively addressed Reviewer 3's concerns, some of which are misunderstandings and not valid criticisms.

Specifically:

Point 1. Estrogen rich media is not the same as the experimental conditions in the paper. The comparison between hormone deprived and short-term estrogen treated conditions allows for a direct assessment of the consequences of estrogen stimulation, something that cannot be achieved in complete media. The authors have addressed this point.

Point 2: there is no expectation or gold standard for spike-in experiments and in our experience, these introduce more errors than they address, in large part because they require physically adding an external variable and assuming the exact same amount is added. The fact that the reviewers did this experiment is commendable but not necessary in my opinion. The other subpoints have been addressed.

Point 3: The extra text clarifies this. The second point is a misunderstanding. It's not 72h vs 48h of stimulation, it's 72h of estrogen deprivation, so this has now been addressed.

All editorial and formatting issues were resolved by the authors.

Dear Dr. Notani,

I am pleased to inform you that your manuscript has been accepted for publication in the EMBO Journal.

Yours sincerely,

Cornelius Schneider, PhD
Editor
The EMBO Journal
c.schneider@embojournal.org
